# Engineering artificial photosynthetic life-forms through endosymbiosis

Jay Cournoyer [1], Sarah D. Altman[1], Yang-le Gao[1], Catherine L. Wallace [2], Dianwen Zhang[2], Guo-Hsuen Lo [1], Noah T. Haskin[1] & Angad P. Mehta [1]✉

The evolutionary origin of the photosynthetic eukaryotes drastically altered the evolution of complex lifeforms and impacted global ecology. The endosymbiotic theory suggests that photosynthetic eukaryotes evolved due to endosymbiosis between non-photosynthetic eukaryotic host cells and photosynthetic cyanobacterial or algal endosymbionts. The photosynthetic endosymbionts, propagating within the cytoplasm of the host cells, evolved, and eventually transformed into chloroplasts. Despite the fundamental importance of this evolutionary event, we have minimal understanding of this remarkable evolutionary transformation. Here, we design and engineer artificial, genetically tractable, photosynthetic endosymbiosis between photosynthetic cyanobacteria and budding yeasts. We engineer various mutants of model photosynthetic cyanobacteria as endosymbionts within yeast cells where, the engineered cyanobacteria perform bioenergetic functions to support the growth of yeast cells under defined photosynthetic conditions. We anticipate that these genetically tractable endosymbiotic platforms can be used for evolutionary studies, particularly related to organelle evolution, and also for synthetic biology applications.

[1] Department of Chemistry, University of Illinois at Urbana-Champaign, 600 S Mathews Avenue, Urbana, IL 61801, USA. [2] The Imaging Technology Group, Beckman Institute for Advanced Science & Technology, University of Illinois at Urbana-Champaign, 405 North Mathews Avenue, Urbana, IL 61801, USA. ✉email: apm8@illinois.edu

The origin of eukaryotic cells is a fundamental milestone in the evolution of complex life forms, and the evolution of organelles is one of the key steps in eukaryogenesis. Based on the endosymbiotic theory, eukaryotic organelles, like mitochondria and chloroplasts, are proposed to have originated and evolved from bacterial endosymbionts during an early stage of evolution (Fig. 1A)[1–5]. This momentous evolutionary event resulted in the origin of eukaryotic cells, followed by subsequent evolution of versatile life forms that significantly impacted global ecology and ecosystems. Since organelles like mitochondria and chloroplasts possess their own genomes, DNA sequencing and biochemical studies spanning several decades have supported this hypothesis[3,6]. Despite the significant importance of endosymbiosis to the evolution of life and global ecology, we have little idea of how bacterial endosymbionts were established within host cells and how they evolved and transformed into organelles. Some of the key questions central to the transformation of bacterial endosymbionts into organelles are: (i) what are the minimal factors and mechanisms necessary for establishing endosymbiosis? (ii) how did endosymbiont genome minimization occur? (iii) how did the endosymbiont metabolism evolve once in a host cell?

It has been a long-standing quest to identify and characterize naturally existing endosymbiotic systems[7,8] for evolutionary studies and synthetic applications. Particularly for photosynthetic endosymbiosis, as early as 1930s, artificial endosymbiosis has

been attempted with photosynthetic algae and freshly extracted mammalian cells[9,10]. Several subsequent studies have attempted to transiently propagate photosynthetic bacteria, algae or photosynthetic organelles within eukaryotic cells like zebrafish embryonic cells and animal cells[11,12]. While these systems suggest the possibility of a photosynthetic bacterial cell to survive within the eukaryotic cells, none of the photosynthetic bacteria were able to fully support the host bioenergetic functions or carbon-source requirements similar to chloroplasts in algal and plant cells. We believe that building genetically tractable model endosymbiotic systems that perform organelle-like functions (e.g., ATP synthesis and supply, carbon assimilation) will provide an endosymbiotic platform that can be metabolically manipulated, analytically studied and imaged, and computationally modeled and predicted. Such platforms will break the gridlock on our understanding of the evolutionary origin of photosynthetic eukaryotic cells. We (along with Schultz) had previously engineered *E. coli* endosymbionts within yeast cells to rescue compromised mitochondrial function[13]. This is a genetically tractable platform that can be used to recapitulate mitochondrial evolution[14].

Inspired by the evolutionary observations and previous synthetic efforts, we sought to engineer genetically tractable platforms where the endosymbiotic, photosynthetic bacteria perform chloroplast-like functions. To design a platform for artificial photosynthetic endosymbiosis, we used key observations from

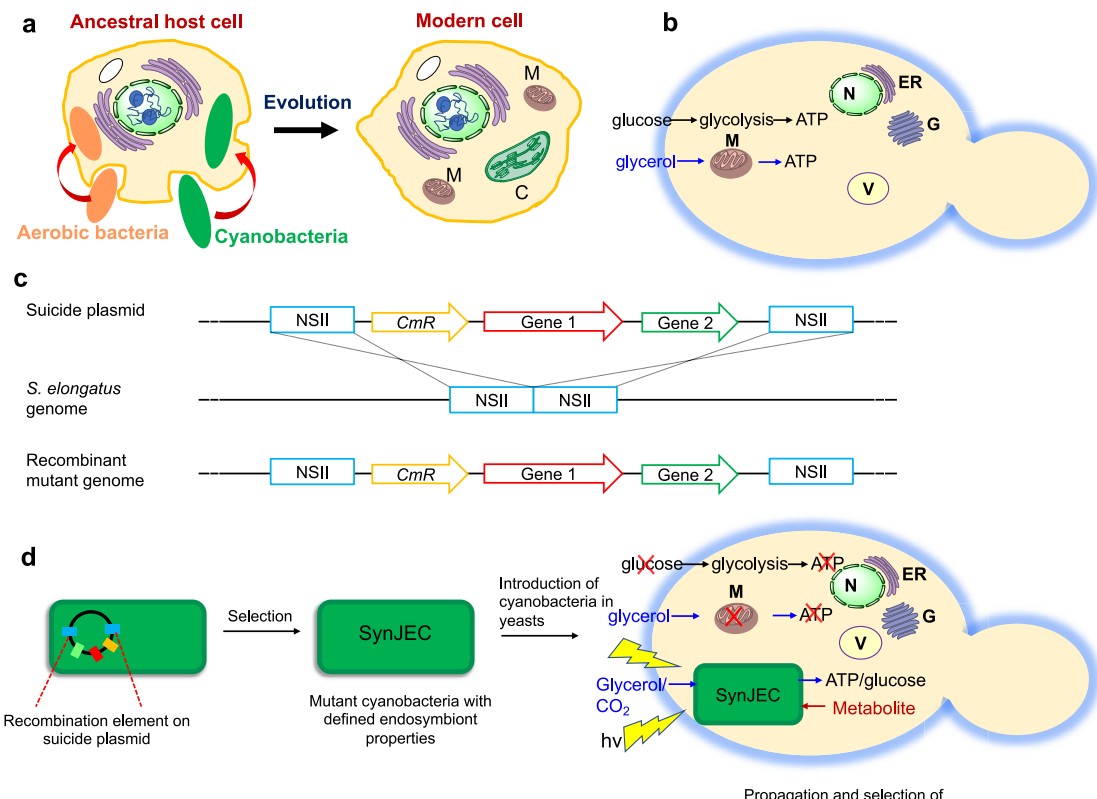

**Fig. 1 Endosymbiotic theory and our platform to recapitulate the evolution of photosynthetic eukaryotic cells. a** The endosymbiotic theory–Mitochondria, M, are proposed to have evolved from a class of α-proteobacteria while the chloroplast, C, is proposed to have originated from cyanobacteria. Golgi apparatus —G, Endoplasmic reticulum—ER, Vacuole—V. **b** *S. cerevisiae* (budding yeast) cells produce ATP by glycolysis or mediated oxidative phosphorylation. **c** Suicide plasmid-based strategy used in this manuscript to engineer cyanobacterial mutants, SynJEC strains. **d** Our platform: We use suicide plasmid-based strategy to engineer cyanobacterial endosymbionts, SynJEC strains, such that they perform chloroplast-like functions. *S. cerevisiae* mutants, deficient in ATP synthesis by oxidative phosphorylation under defined photosynthetic selection conditions, are used as host strains. Engineered cyanobacteria strains, SynJEC, are then introduced into the yeast cells by a cell fusion process that is developed and optimized (see Methods). The yeast/cyanobacterial chimera are selected under defined photosynthetic selection conditions where the cyanobacterial endosymbionts provide ATP to the mutant *S. cerevisiae* host cells, and *S. cerevisiae* provide essential metabolites to the *S. elongatus* endosymbionts.

studies related to chloroplast evolution. First, we had to identify and engineer genetically tractable photosynthetic bacteria such that they could perform chloroplast-like functions within the cytoplasm of the host cell. In case of chloroplast evolution, there is a consensus that chloroplasts evolved from cyanobacterial endosymbionts; several sequencing and biochemical studies support this proposal. Based on this evidence, it is suggested that *Gloeomargarita* are the closest identified relatives of chloroplasts[15,16]. While strains of *Gloeomargarita* are not readily genetically manipulable, genetic tools have been developed to manipulate several strains of *Synechococcus*, which are relatives of *Gloeomargarita*[17,18]. The exact nature of the host cell is still elusive[19], and it has been suggested that photosynthetic endosymbionts were established in various eukaryotic host cells during the evolution of chloroplasts[4,20]. Because of these observations, for our engineering efforts, we decided to engineer artificial endosymbiosis between genetically tractable *Synechococcus elongatus* PCC 7942 (Syn7942) and a model eukaryotic cell, *Saccharomyces cerevisiae*, budding yeast. Next, to determine the role that Syn7942 would play in our synthetic endosymbiotic system, we again sought inspiration from chloroplast function and evolution. Though the modern-day chloroplasts perform various functions like ATP synthesis, carbon assimilation, sulfate assimilation, nitrate assimilation, amino acid biosynthesis, fatty acids biosynthesis amongst others[4], the key drivers of endosymbiotic evolution of chloroplast are still unclear due to extensive horizontal gene-transfer, secondary endosymbiosis and endosymbiotic adaptation[21]. However, it is widely suggested that bioenergetic considerations may have been the key drivers of organelle evolution[21,22]. This is highlighted by the fact that ATP/ADP translocases and transporters are widely conserved across organelles like mitochondria and chloroplasts, including organisms that are related to the endosymbiotic precursors of mitochondria and chloroplast[23,24]. These observations were central in designing and engineering artificial photosynthetic endosymbiosis between model yeasts and model cyanobacteria.

In this work, we engineer an endosymbiotic platform such that the mutant cyanobacterial endosymbionts perform bioenergetic function for the host cells, i.e., endosymbionts provide ATP generated through photophosphorylation to the host cells (Fig. 1b–d). Through series of cyanobacterial and yeast engineering efforts we are able to establish artificial photosynthetic endosymbiosis between yeast mutants and cyanobacterial mutants to generate yeast/cyanobacteria chimeras that are able to propagate through at least 15 to 20 generations of growth under optimal photosynthetic growth conditions. The yeast/cyanobacteria chimeras are characterized by analyzing their viability under photosynthetic selection conditions, analysis of the total genomic DNA, metabolic coupling of endosymbiont/host, dependence on photosynthesis, pseudo-total internal reflection fluorescence (pTIRF) microscopy, fluorescence confocal microscopy and transmission electron microscopy (TEM). Our studies also highlight the critical genetic elements that allow us to engineer artificial photosynthetic endosymbiosis between model cyanobacteria and model eukaryotic cells. We anticipate that such genetically tractable photosynthetic platforms will have significant implications on synthetic biology applications[25–27]. Further, these photosynthetic endosymbiotic systems could also provide a platform to recapitulate various evolutionary trajectories related to the conversion of photosynthetic endosymbionts into photosynthetic organelles (i.e., chloroplasts).

## Results

### Engineering Syn7942 to function as ATP-providing endosymbionts within yeast cells.
Phylogenetic analysis of nucleotide transport proteins (NTT) suggests that cyanobacteria likely have uncharacterized nucleotide transport fusion proteins (similar to ADP/ATP translocases)[24]. However, chloroplasts are thought to possess NTTs that possibly originated from intracellular organisms such as chlamydial[23,24]. Therefore, to emulate bioenergetic functions of the chloroplast, our goal was to engineer Syn7942 to efficiently export ATP in the presence of ADP by expression of previously characterized, recombinant ADP/ATP translocase[28–30]. We began by generating a control mutant (SynJEC0) which was obtained by a transformation and selection process developed by Golden and coworkers[18]. Syn7942 was transformed with a plasmid, pCV0055, containing a chloramphenicol resistance cassette placed between two homologous recombination domains, known as neutral sites (NS) in the Syn7942 genome[17], and the mutant strain SynJEC0 was selected on BG-11 selection medium containing chloramphenicol. The generation of this mutant was confirmed by PCR analysis, demonstrating the presence of gene(s) integrated into the Syn7942 genome at the NSII site while the vector was eliminated (Supplementary Fig. 2). Having generated control SynJEC0 mutant strain, we then engineered Syn7942 mutants expressing ADP/ATP translocase from intracellular organisms. Starting with pCV0055, we constructed a plasmid (pML3) (Supplementary Fig. 1 for plasmid map) encoding: (i) codon-optimized ADP/ATP translocase gene, *ntt1*, from *Protochlamydia amoebophila* UWE25[28], a bacterial endosymbiont of *Acanthamoeba spp*, (ii) a chloramphenicol selection marker and (iii) NSII homology region for genomic recombination. We transformed Syn7942 with pML3 to generate the mutant strain SynJEC1. To determine if the *ntt1* was functional when expressed in SynJEC1, we assayed the ADP/ATP translocase activity (using luciferase assays) with SynJEC1 cells and compared it to the SynJEC0 control mutant. We observed that SynJEC1 released a significant amount of ATP when compared to control cells to which ADP was not added, and also as compared to the control strain SynJEC0 (Fig. 2a) to which ADP was added. However, unlike *E. coli* cells[13,28], we observed substantial amount of ATP released when ADP was added to the SynJEC0 cells that did not contain ADP/ATP translocase gene from *Protochlamydia amoebophila* UWE25 (Fig. 2a and Supplementary Fig. 7). This observation suggests the possibility of ADP/ATP translocase-like proteins that might be already encoded by the Syn7942 genome itself, consistent with previous phylogenetic diversity analysis of NTTs[24].

### Generation and selection of the *S. elongatus/S. cerevisiae* chimera.
To introduce cyanobacterial cells into yeast cells, we optimized a polyethylene glycol (PEG)-induced fusion and selection protocol which was used previously to install mitochondria[31] and *E. coli* cells[13] in yeast spheroplasts. For host cells, we used *S. cerevisiae cox2-60*, a strain that is incapable of assembling a functional cytochrome c oxidase complex and consequently has a respiration-deficient phenotype specifically due to lack of ATP synthesis under defined selection conditions[32–35]. Under respiration selection conditions, the rescue of this phenotype can be observed by restoring ATP synthesis using plasmid (pML64) based expression of COX2 gene (Fig. 2e) as described previously[35]. Our calculations suggest that the doubling time of *S. cerevisiae cox2*-60-pML64 is around 7 h (Supplementary Fig. 8). We next evaluated if ATP providing cyanobacterial endosymbionts could similarly rescue the phenotype of *S. cerevisiae cox2-60* under photosynthetic selection conditions. We fused SynJEC0 and SynJEC1 to the *S. cerevisiae cox2-60* cells and selected fusions by growing mixtures on partial selection conditions containing non-fermentable carbon source and low levels of fermentable carbon source (1% yeast extract, 2%

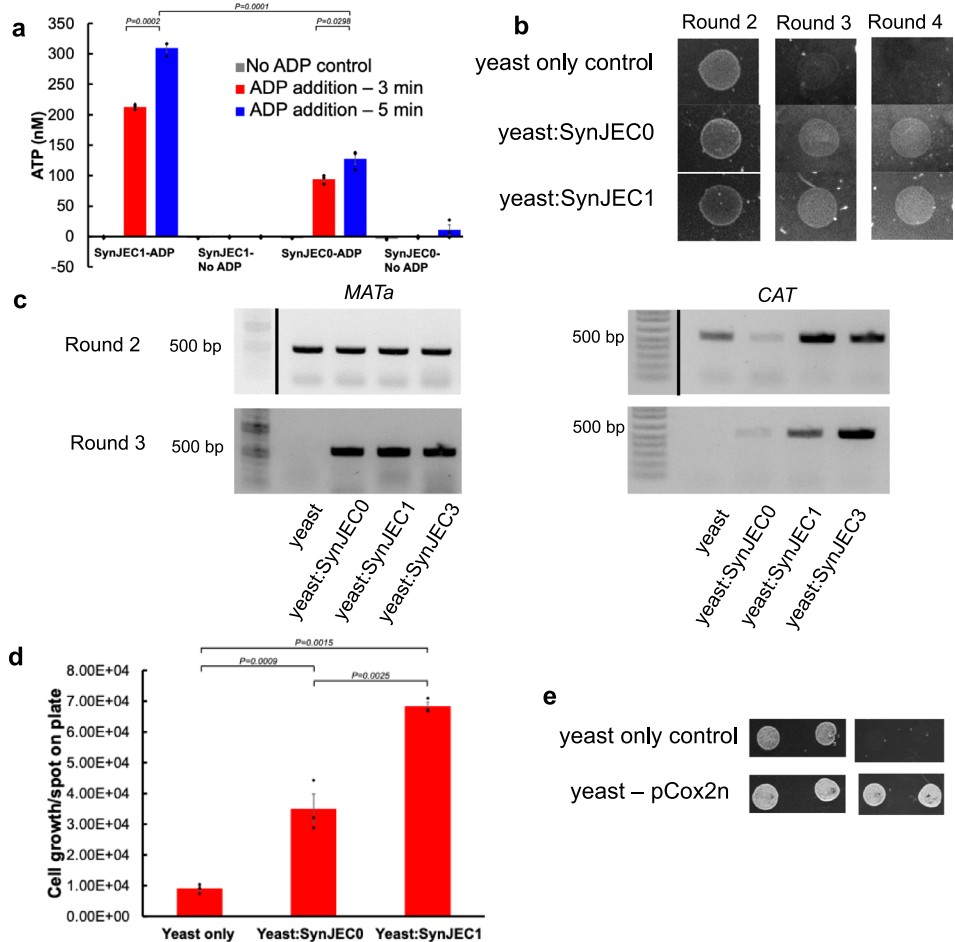

**Fig. 2 S. cerevisiae–SynJEC chimeras have a partially rescued respiration-competent phenotype. a** Release of ATP by *SynJEC1* cells expressing the *UWE25* ADP/ATP translocase in the presence of 80 μM ADP in comparison to *SynJEC0* cells. ATP was released when SynJEC1 (expressing the ATP/ADP translocase) and SynJEC0 cells were challenged with extracellular ADP (80 μM), but not with a blank solution lacking ADP (*N* = 3 biological replicates; data are presented as mean values + /− SEM). Two-sided *t*-tests were used to compare means without adjustments (95% CI, Cohen's *d* = 10.6, DF = 4, *P* = 0.0002; 95% CI, Cohen's *d* = 13.0, DF = 4, *P* = 0.0001; 95% CI, Cohen's *d* = 2.7, DF = 4, *P* = 0.0002). **b** Growth of *S. cerevisiae* cox2-60–*SynJEC* chimeras on medium containing glycerol as the sole carbon source. No growth was observed in round 4 for yeast lacking intracellular *SynJEC*. The experiment was repeated independently six times with similar results. **c** Total DNA isolated from spots grown on selection medium III contain the yeast-encoded *MATa* gene and *SynJEC*-encoded chloramphenicol acetyltransferase (*CAT*) gene. The experiment was repeated independently six times with similar results. **d** Growth rate of yeast-*SynJEC* chimeras on Selection Medium III. Cells (3.00 ×10³) were spotted on Selection Medium III and counted after 72 h growth. (*N* = 3 technical replicates; data are presented as mean values+/− SEM.)*P*-values were calculated by two-tailed *t*-test comparing the two means. **e** Panel 1 describes the growth of *S. cerevisiae-cox2-60* and *S. cerevisiae-cox2-60*-pCOX2n under non-selection conditions. Panel 2 describes the growth of *S. cerevisiae-cox2-60* and *S. cerevisiae-cox2-60*-pCOX2n under selection conditions where the rescue in the growth of *S. cerevisiae-cox2-60*-pCOX2n is observed but no growth is observed for *S. cerevisiae-cox2-60*. The experiment was repeated independently three times with similar results. Source data are provided as a Source Data file.

peptone, 1 M sorbitol, 3% glycerol, 0.1 % glucose, 1X BG-11; selection medium I) in presence of light. The fusions were propagated in 12 h light-dark cycles at 30 °C. Under these conditions, we did not observe yeast colonies with control *S. cerevisiae cox2-60* cells but did observe small, distinct colonies for *S. cerevisiae cox2-60*-SynJEC0 and *S. cerevisiae cox2-60*-SynJEC1 fusions. Colonies were picked and re-plated for four consecutive rounds of regrowth: one round on selection medium II (1% yeast extract, 2% peptone, 1 M sorbitol, 3 % glycerol, 0.1% glucose, 1X BG-11, 50 mg/ml carbenicillin; selection medium II) and two to three rounds on selection medium III (1% yeast extract, 2% peptone, 1 M sorbitol, 3 % glycerol, 50 mg/ml carbenicillin, 1X BG-11; selection medium III) (Fig. 2b). Note, in selection medium II and III, carbenicillin was added to eliminate any extracellular cyanobacteria. As expected, *S. cerevisiae cox2-60* cells failed to grow on selection medium III during subsequent rounds of regrowth

when they were not fused to Syn7942. We observed higher growth for *S. cerevisiae cox2-60*-SynJEC0 chimeras as compared to the host strain by itself (i.e., *S. cerevisiae cox2-60*) on selection medium III through consecutive rounds of re-plating (Fig. 2b). To calculate the doubling numbers in case of each of the fusions, we plated a defined number of starting cells and determined the endpoint cell-count. We observed that the host *S. cerevisiae cox2-60* propagated for only ~ 2 doublings on selection medium III and *S. cerevisiae cox2-60*-SynJEC0 chimeras propagated for ~10 doublings on selection medium III (Fig. 2D). We observed robust phenotypic rescue for *S. cerevisiae cox2-60*-SynJEC1 (Fig. 2b, d); the *S. cerevisiae cox2-60*-SynJEC1 continued to propagate for ~14 doublings (Supplementary Table 5) followed by a drop in the growth rate on selection medium III. These data suggested that the expression of the ADP/ATP translocase was important in restoring respiration competency in *S. cerevisiae*

**Table 1 A list of cyanobacterial strains engineered and used in this study.**

| Strain designation | Suicide plasmid | Plasmid described | Genotype |
|---|---|---|---|
| SynJEC0 | pCV0055 | Golden et al.[18] | NSII::CAT |
| SynJEC1 | pML3 | This study | NSII::CAT, UWE25-ntt1 |
| SynJEC2 | pML14 | This study | NSII::CAT, UWE25-ntt1, Ctr-incA |
| SynJEC3 | pML17 | This study | NSII::CAT, UWE25-ntt1, Ctr-incA, CT_813 |
| SynJEC4 | pML28 | This study | NSII::CAT, UWE25-ntt1, Ctr-incA, CT_813, Cca-incA |
| SynJEC5 | pML14 and pML62 | This study | NSII::CAT, UWE25-ntt1, Ctr-incA, CT_813; metA::KanR |
| SynJEC9 | pML3 and pML58 | This study | NSII::CAT, UWE25-ntt1; metA::KanR |

Note: See Supplementary Fig. 1 and Supplementary Table 3 for detailed plasmid maps.

*cox2-60*/cyanobacteria chimera. This phenotypic rescue of the host, *S. cerevisiae cox2-60,* indicated that the cyanobacterial endosymbionts, especially SynJEC1, could partially rescue the growth of the host *S. cerevisiae cox2-60* cells under selection conditions. It is possible that a weaker phenotypic rescue of the *S. cerevisiae cox2-60* cells when fused with SynJEC0 could be due to the background ADP/ATP translocase activity of SynJEC0 cells that we detected during our translocase activity studies described above. Our observations are consistent with the studies suggesting that cyanobacterial endosymbionts may have needed exogenous translocases to facilitate their bioenergetic functions[23,24].

To characterize the presence of cyanobacterial endosymbionts within yeast cells, we isolated total genomic DNA from fused yeast cells propagated for multiple generations under selection growth conditions to eliminate all the extracellular cyanobacteria (if any, see Supplementary Fig. 4, Supplementary Fig. 6), and performed PCR analysis to determine the presence of both yeast and cyanobacterial genomes. We detected the presence of both the yeast *MATa* gene and mutant Syn7942 chloramphenicol acetyltransferase (*CAT*) gene in the colonies by PCR (Fig. 2c), suggesting the presence of both yeast and cyanobacterial genomes. Interestingly, this set of experiments suggested that unlike the engineered yeast/*E. coli* endosymbiosis studies[13], Syn7942 cells expressing ADP/ATP translocase were sufficient to transiently recover the growth phenotype of *S. cerevisiae cox2-60* under selection conditions. In addition, PCR analysis for cyanobacterial genomes suggest that SynJEC0 cells that did not contain the ADP/ATP translocase gene, were also able to establish transient endosymbiosis with *S. cerevisiae cox2-60.* Again these results were unlike the yeast/*E. coli* endosymbiosis studies[13] where it was necessary to express ADP/ATP translocase and SNARE-like proteins for establishing endosymbiosis with yeast cells. This set of experiments suggested therefore that cyanobacterial relatives of chloroplast precursors could have possessed innate ability to establish bioenergetically relevant endosymbiosis; however, as predicted[23,24], the cyanobacterial endosymbionts might have needed exogenous translocases to facilitate robust endosymbiosis.

**Improving the stability and growth of yeast/cyanobacteria chimera.** It is hypothesized that the endosymbionts could have acquired a variety of genes through horizontal gene transfer which facilitated the establishment of robust endosymbiosis within the cytoplasm of the host cells[15,36]. Based on the previous observations with synthetic endosymbiotic systems[13], we hypothesized that we could improve the stability of the cyanobacterial endosymbionts by engineering mechanisms that are known to evade intracellular degradation within the cytoplasm of the host cells. Literature suggests that one mechanism by which intracellular pathogenic bacteria avoid intracellular degradation is through the expression of SNARE-like proteins[37,38], which are thought to inhibit SNARE-mediated membrane fusion through mimicry of the host SNAREs. Our previous studies suggested that expression of *Chlamydia trachomatis* genes *incA* and *CT_813* along with the *Chlamydia caviae* gene *incA* improved the stability of *E. coli* endosymbionts within yeast cells[13]. Therefore, we evaluated if expressing a combination of SNARE-like proteins along with ADP/ATP translocase improves the stability of yeast/cyanobacteria chimeras.

To this end, we began by constructing a series of integrative suicide plasmids (pML14, pML17, pML28) starting from pCV0055; in addition to recombination/selection elements, each of these plasmids contained ADP/ATP translocase and a combination of gene fragments corresponding to one or more SNARE-like proteins driven by a promoter, Ptrc. Through transformation, recombination and selection methods we generated a series of *S. elongatus* mutant strains described in Table 1 (SynJEC2, SynJEC3, SynJEC4), each of which expressed ADP/ATP translocase and one or multiple SNARE-like proteins. Each of these mutants were confirmed by isolation of the genomic DNA, PCR amplification of the mutant locus and sequencing of the corresponding mutant locus (Supplementary Fig. 2). We then individually fused SynJEC2, SynJEC3, SynJEC4 to *S. cerevisiae cox2-60* to determine if the expression of SNARE-like proteins improved the stability of the yeast/cyanobacteria chimeras. As before, colonies were picked and continuously re-plated for four consecutive rounds of regrowth: two rounds on selection medium II and two rounds on selection medium III to identify optimal endosymbiosis that would result in higher stability and more generations of growth. Through this experiment, we found that *S. cerevisiae cox2-60*-SynJEC4 chimeras went extinct in the shortest amount of time compared to the *S. cerevisiae cox2-60*-SynJEC2 and *S. cerevisiae cox2-60*-SynJEC3 lineages (Fig. 3a, c, d). The *S. cerevisiae cox2-60*-SynJEC3 chimeras were the longest-lived; in our hands, we typically observed around 20 doublings on selection medium III (Supplementary Table 5). As per our genomic DNA analysis of the chimeras, after more than 20 rounds of doublings, we observe that the chimeras lose their viability possibly due to loss of cyanobacteria (Supplementary Fig. 11); our imaging studies described below confirm this observation (Supplementary Fig. 4). Our data suggests that the doubling times of *S. cerevisiae cox2-60*-SynJEC3 chimeras is around 10 h, possibly due to the slow growth rate of *S. elongatus strains* (doubling time ~6 to 8 h under optimal media conditions). Further our data indicate that during a single round or re-growth, 3000 cells of *S. cerevisiae cox2-60*-SynJEC3 chimera propagated to $3 \times 10^5$ cells under stringent selection conditions in 72 h, whereas *S. cerevisiae cox2-60* cells are not viable at all under these conditions. This suggests that cyanobacteria are metabolically active and are able to provide ATP to the host cox2-60 strain; and hence endosymbiotic. Taken together, these data suggested that both the ADP/ATP translocase and *C. tr.* IncA and CT_813 SNARE-like proteins play a beneficial role for the

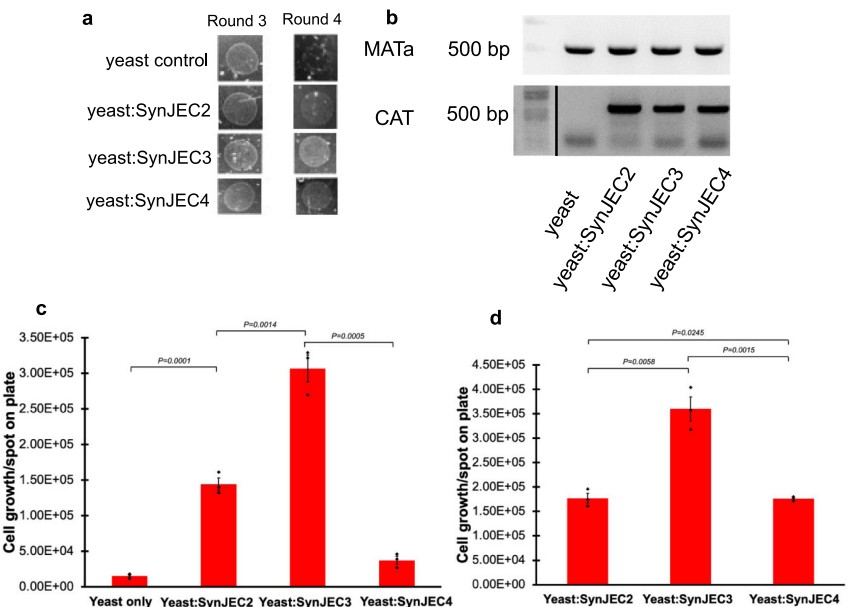

**Fig. 3 Rescue of respiration deficient phenotype by yeast-*SynJEC* chimeras expressing an ATP/ADP translocase and SNARE-like proteins. a** Growth of *S. cerevisiae* cox2-60–*SynJEC2—4* chimeras on medium containing glycerol as the sole carbon source. The experiment was repeated three times independently with similar results. **b** Total DNA of yeast-*SynJEC* chimeras contains yeast *MATa* and *SynJEC CAT* genes. The experiment was repeated three times independently with similar results. **c**, **d** Growth trends of *S. cerevisiae* cox2-60 (yeast only strain), *S. cerevisiae* cox2-60- SynJEC2 (yeast-SynJEC2), *S. cerevisiae* cox2-60- SynJEC3 (yeast-SynJEC3) and *S. cerevisiae* cox2-60- SynJEC4 (yeast-SynJEC4) chimeras on Selection Medium III. Cells ($3.00 \times 10^3$) were spotted on Selection Medium III the final number of cells/spot on plate were determined after 72 h of growth ($N = 3$ technical replicates; data are presented as mean values+/− SEM). *P*-values were calculated by two-tailed *t*-test comparing the two means. Source data are provided as a Source Data file.

cyanobacterial endosymbionts. In each of the above cases, we isolated total genomic DNA from fused yeast cells propagated for multiple generations under selection growth conditions (Supplementary Fig. 6) and PCR analyzed the presence of both yeast and cyanobacterial genomes, and as expected, we detected the presence of both yeast *MATa* and cyanobacterial *CAT* genes (Fig. 3b).

**Imaging cyanobacterial endosymbionts within yeast cells by total internal reflection fluorescence microscopy**. The phenotypic rescue of *S. cerevisiae* cox2-60 and PCR analysis of total genomic DNA from our fusions suggested the presence of cyanobacterial endosymbionts present within yeast cells. To obtain more evidence supporting this possibility, we used fluorescence microscopy approaches detect the presence of cyanobacterial endosymbionts within yeast cells. Because Syn7942 cells possess natural fluorescence properties due to the presence of phycobilins and chlorophyll, we characterized our endosymbiotic chimeras using a home-built total internal reflection fluorescence (TIRF) microscope. In order to enable deep imaging inside live cells, the microscope was used in pTIRF mode, where the incident angle of the laser beam is tuned to be just below the critical angle at the top surface of coverslip, so the laser can travel through yeast cell wall and illuminate above the penetration depth of the excitation laser evanescent field. The microscope also has a Zeiss condenser for brightfield microscopy with white light illumination. In the experiment, yeast cells were first found and imaged in brightfield mode and then pTIRF mode was used to check and detect cyanobacterial autofluorescence. First, we imaged live Syn7942 cells with the TIRF microscope, and as expected, we detected the natural cyanobacterial fluorescence corresponding to the emission (661 nm) of phycobilins and chlorophyll[39,40] in cyanobacteria using 561 nm laser excitation (Fig. 4a). Under the same conditions, no fluorescent signals were detected for the control *S.*

*cerevisiae* cox2-60 host cells (Fig. 4a). This observation suggested that we should be able to detect the presence of Syn7942 derived cells within *S. cerevisiae* cox2-60 with the TIRF microscopy without need for the expression of heterologous fluorescent entities. During each round of re-plating under selection conditions, we used the TIRF microscopy to image live chimeras. As shown in Fig. 4a we observed prominent punctate signals corresponding to cyanobacteria in case of all the fusions that demonstrated phenotypic rescue of the host yeast cells, i.e., *S. cerevisiae* cox2-60-SynJEC0, *S. cerevisiae* cox2-60-SynJEC1 and *S. cerevisiae* cox2-60-SynJEC3 chimeras.

**Imaging cyanobacterial endosymbionts by confocal fluorescence microscopy**. To obtain a better understanding of the fraction of yeast cells containing cyanobacterial endosymbionts, we next used confocal fluorescence microcopy approaches which allows us to precisely measure endosymbiont distribution on fixed cells. We first propagated the *S. cerevisiae* cox2-60-SynJEC3 chimeras for one round of growth on selection medium II and one round of growth on selection medium III. Following this selection, we confirmed the detection of cyanobacterial endosymbionts within yeast cells by PCR analysis of the total genomic DNA and pTIRF microscopy. Following this confirmation, the *S. cerevisiae* cox2-60-SynJEC3 cells were fixed with Karnovsky fixative[41], and stained with FITC-labeled concanavalin A (Con A-FITC). Similarly, the control yeast cells, *S. cerevisiae* cox2-60, were also fixed and stained with Con A-FITC. Samples were then analyzed by using a commercial Leica SP8 fluorescence confocal microscope. In the microscope, a 488 nm laser was used to excite Con A-FITC stained yeast while a 561 nm laser for naturally fluorescent cyanobacterial cells. The spectral detector in the microscope allows up to 3 spectral detection bands, each of which is independent from the others and continuously tunable from 400 nm to 800 nm. Two detection bands with an identical

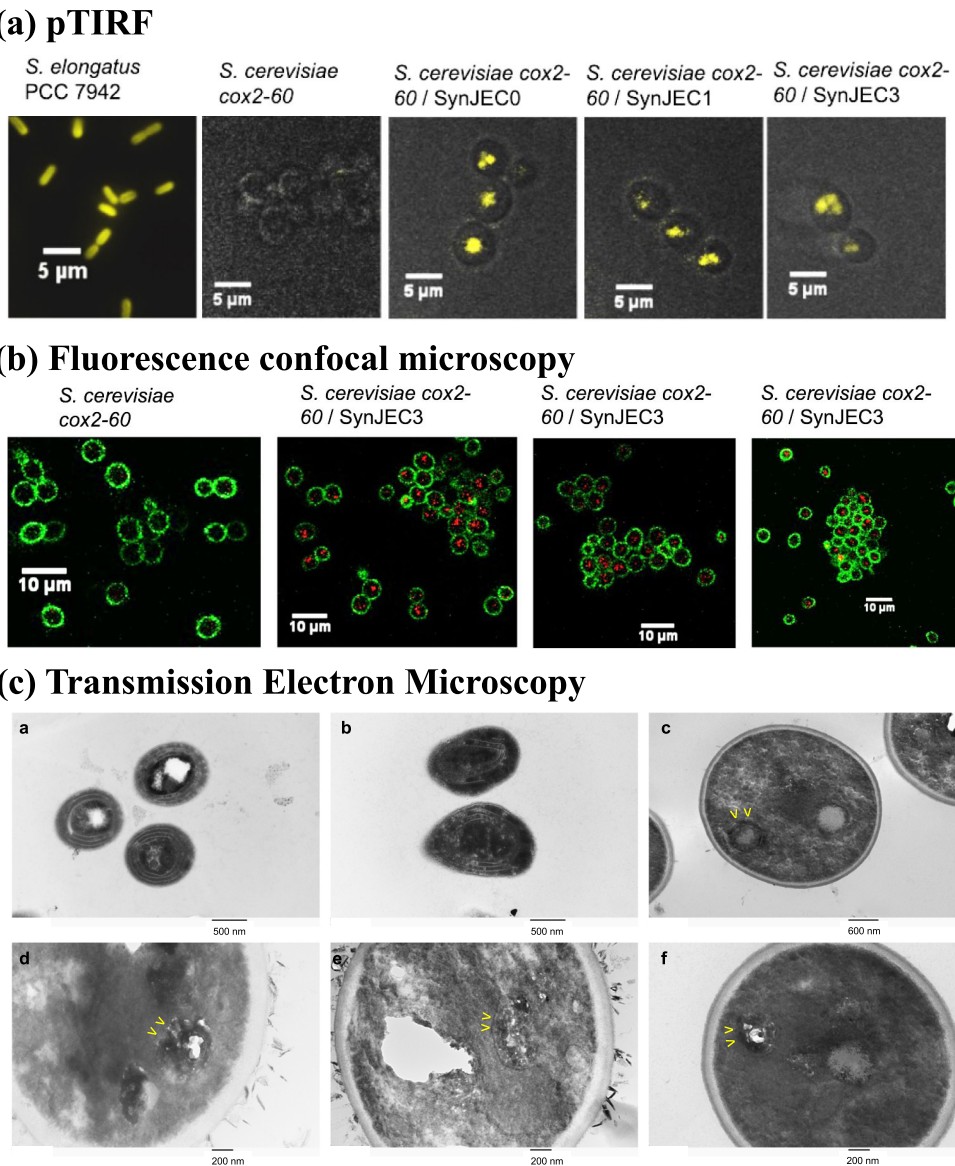

**Fig. 4 Imaging intracellular endosymbiont *Synechococcus* by fluorescent microscopy. a** pTIRF microscopic images of *Synechococcus* cells, control yeast cells, and chimeric cells that were grown under selection conditions (Ex. = 561 nm; Em. = 653/95 nm). Panels are merged images of pTIRF (yellow) and brightfield microscopy (gray). The experiment was repeated three times independently with similar results. **b** Fluorescence confocal microscopy images of control yeast cells and chimeric cells, which were grown under selection conditions. The yeast cell wall was stained with Con A-FITC (green, Ex. = 488 nm; Em. = 510/20 nm) and the presence of cyanobacteria was monitored by cyanobacterial fluorescence (red, Ex. = 561 nm; Em. = 650/20 nm). Based on these images, it is possible that multiple cyanobacterial cells could be present in some of the yeast cells. The experiment was repeated three times independently with similar results. **c** Panel **a** and **b** are cyanobacterial samples imaged by Transmission Electron Microscopy (TEM) and Panels **c**–**f** are fusions imaged by TEM. Yellow arrows show characteristic cyanobacterial structures within the cytoplasm of the yeast cells. The experiment was repeated twice independently with similar results.

bandwidth of 20 nm were set at 510 nm and 650 nm respectively for the green fluorescent light from the Con A-FITC dyes on yeast cells and the red autofluorescence from cyanobacterial cells. Both yeast cells and cyanobacterial cells could therefore be spectrally separated and imaged simultaneously (see Supplementary Fig. 3). As shown in Fig. 4b, we detected the presence of cyanobacterial endosymbionts within the host yeast cells in confocal fluorescence microscopy further confirming the generation, selection and propagation of cyanobacterial endosymbionts within yeast cells after multiple rounds of growth under stringent selection conditions. In order to analyze the distribution of the cyanobacterial endosymbionts within yeast cells, we scanned a large number of cells by confocal microscopy. As shown in Supplementary Fig. 4,

under the fusion, selection and detection conditions a significantly high fraction of yeast cells show signals specific to cyanobacteria in the early stages of chimera propagation. These observations were again fundamentally different from our previous investigations with the yeast/*E. coli* endosymbiotic platform[13].

**Imaging cyanobacterial endosymbionts by transmission electron microscopy.** To further demonstrate the presence of intact cyanobacterial endosymbionts and to understand the localization of endosymbionts within host cells use analyzed the yeast/cyanobacteria chimera using transmission electron microscopy

(TEM). We first analyze SynJEC3 strains by TEM; characteristic cyanobacteria TEM images are shown in Fig. 4c, panels a–b. Next, we generated and propagated *S. cerevisiae cox2-60*-SynJEC3 chimeras for one round of growth on selection medium II and one round of growth on selection medium III (at least 10 doubling detected by yeast cell count analysis). These samples were then fixed, treated and analyzed by TEM as described in the methods section. As shown in Fig. 4c, panels c–f, we observe characteristic structures corresponding to cyanobacteria within the yeast cells. In addition to this, the cyanobacteria appear to be structurally intact and do not appear to be entrapped in any intracellular organelle within yeast cells.

**Metabolic coupling of the yeast/cyanobacterial endosymbiotic system.** The endosymbiotic theory suggests that the host and the endosymbiont evolved metabolic interdependency which was crucial for retaining and optimizing the presence of endosymbionts within the host cells. These features are central to present-day organelle function; for example, organelle membranes possess specific or non-specific transporters that facilitate the uptake of necessary nucleotides, amino acids or amino acid precursors from the host cytosol[42]. Such uptake mechanisms are crucial for organelle genome replication and organelle protein synthesis. Therefore, as a next step, we generated an endosymbiotic system between *S. cerevisiae cox2-60* and *S. elongatus* where the *S. cerevisiae cox2-60* cells depended on *S. elongatus* endosymbionts for ATP and the *S. elongatus* endosymbionts depended on *S. cerevisiae cox2-60* for an essential amino acid.

Analysis of the Syn7942 genome suggests that this strain has several putative amino acid and cofactor transporters. Gene deletion of a key biosynthetic gene usually results in the generation of a corresponding auxotrophic strain (e.g., gene deletion of *metA* results in a methionine auxotroph). As a first step towards creating metabolite interdependency, we generated a SynJEC3 methionine auxotrophic mutant to test if it could establish an endosymbiotic relationship with *S. cerevisiae cox2-60*. To this end, we constructed another suicide plasmid pML58 containing homology regions corresponding to Syn7942 *metA* gene and a kanamycin resistance marker. We transformed pML58 into SynJEC3 and selected for cyanobacterial mutants that were able to grow in BG-11 medium in the presence of chloramphenicol, kanamycin and methionine. Individual colonies were grown in liquid selection medium and genomes were isolated and characterized by PCR analysis to confirm recombination at the *metA* locus. We confirmed that characterized colonies were methionine auxotrophs by growing them in the presence and absence of methionine (Fig. 5a, Supplementary Fig. 5a); the corresponding strain was designated as SynJEC5. Next, SynJEC5 was fused to *S. cerevisiae cox2-60* spheroplasts. Colonies were observed on selection medium II and were able to propagate in selection medium III for four rounds of re-plating (Fig. 5b, Supplementary Fig. 5d,e). As before, total genomic DNA was isolated and PCR analysis was performed to confirm the presence of cyanobacterial endosymbionts (Fig. 5c, Supplementary Fig. 5b). pTIRF microscopy further supported the presence of cyanobacterial endosymbionts in yeast cells (Supplementary Fig. 5c).

**Growth dependence of the yeast/cyanobacteria chimeras on photosynthesis.** Because light driven photosynthesis in *S. elongatus* is expected to be crucial towards ATP synthesis and therefore endosymbiosis in our platform, we investigated the importance of photosynthesis to the survival and propagation of yeast/cyanobacteria chimeras. To this end, we compared the growth of chimeras in a day/night cycle to those grown in the absence of light. As shown in Fig. 5e, we saw significantly

retarded growth for *S. cerevisiae cox2-60*-SynJEC3 chimeras when propagated under non-photosynthetic conditions as compared to the photosynthetic conditions; particularly, we just observed two doublings for *S. cerevisiae cox2-60*-SynJEC3 chimeras under non-photosynthetic conditions (identical to *S. cerevisiae cox2-60* alone under identical selection conditions). Because *S. elongatus* strains have an extensive metabolic adaptation and stress response to adapt to dark environment, we also wanted to evaluate the effect of light starvation on cyanobacterial endosymbionts prior to fusion to the yeast cells. When light is no longer available, *S. elongatus* cells catabolize their glycogen stores through respiration and can remain viable for significant periods[43,44]. Therefore, in order to test the effect of light starvation prior to fusion on yeast/cyanobacteria endosymbiosis, we incubated SynJEC3 strains in darkness for 96 h prior to fusion with *S. cerevisiae cox2-60* spheroplasts. The recovered fusions were kept under strict darkness for during all rounds of selection and propagation. In all instances, we observed no growth for *S. cerevisiae cox2-60*-SynJEC3 chimeras generated from this experiment but observed normal growth when chimeras were generated under photosynthetic conditions described earlier (Fig. 5f). This suggested that light is essential for the establishment and propagation of cyanobacterial endosymbionts within the yeast cells. We also tracked the propagation of *S. cerevisiae cox2-60*-SynJEC3 chimeras by pTIRF microscopy to determine the relative changes in the levels of photosynthetic apparatus, particularly phycobilisomes (PBS) and chlorophyll *a* (Chl). As indicated by the Supplementary Fig. 9, we observe alteration of internal ratio of PBS:Chl indicating possible changes to total composition of the photosynthetic apparatus in the endosymbiont over multiple rounds of growth.

Next, we evaluated the effects of photosynthesis inhibitor, 3-(3,4-dichlorophenyl)−1,1-dimethylurea (DCMU) on the viability of yeast/cyanobacteria chimera. We first demonstrated that DCMU completely arrested the growth of cyanobacteria but had no affect the viability of *S. cerevisiae cox2-60* (Supplementary Fig. 10). On the other hand, as expected, the *S. elongatus* strains completely lose their viability on DCMU treatment[45]. When *S. cerevisiae cox2-60*/SynJEC3 were treated with DCMU, we also observed loss of viability (Fig. 5g, h) indicating again, the essential role of endosymbiont mediated photosynthesis on the viability of the yeast/cyanobacteria chimeras.

## Discussion

In this study, our goal was to engineer genetically tractable platforms where the endosymbiotic bacteria perform chloroplast-like functions for the host cell. To build such a system, we were inspired by the evolutionary origin of photosynthetic eukaryotic cells which suggests that photosynthetic eukaryotic cells originated due to endosymbiosis between non-photosynthetic eukaryotic cells and photosynthetic cyanobacterial or algal endosymbionts. The geological record places the advent of the earliest eukaryotes about 2.7–1.7 billion years ago (Gya)[46,47], coinciding with an explosion of cyanobacterial-derived oxygen ($O_2$) levels in the biosphere of the Earth known as the Great Oxidation Event (GOE)[48,49]. The harsh conditions of GOE-era Earth are also thought to be connected to the evolution of bacterial endosymbiont-derived organelles such as mitochondria and chloroplasts. Though primary photosynthetic endosymbiosis was once thought to be an unprecedented event which only occurred 2.7–1.7 Gya, phylogenetic analysis has revealed that the chromatophore of the ameboid *Paulinella*, which contains cyanobacterial endosymbionts, diverged from its relatives about 140 million years ago through a separate primary endosymbiotic event[50–52]. Comparative genomic studies have provided

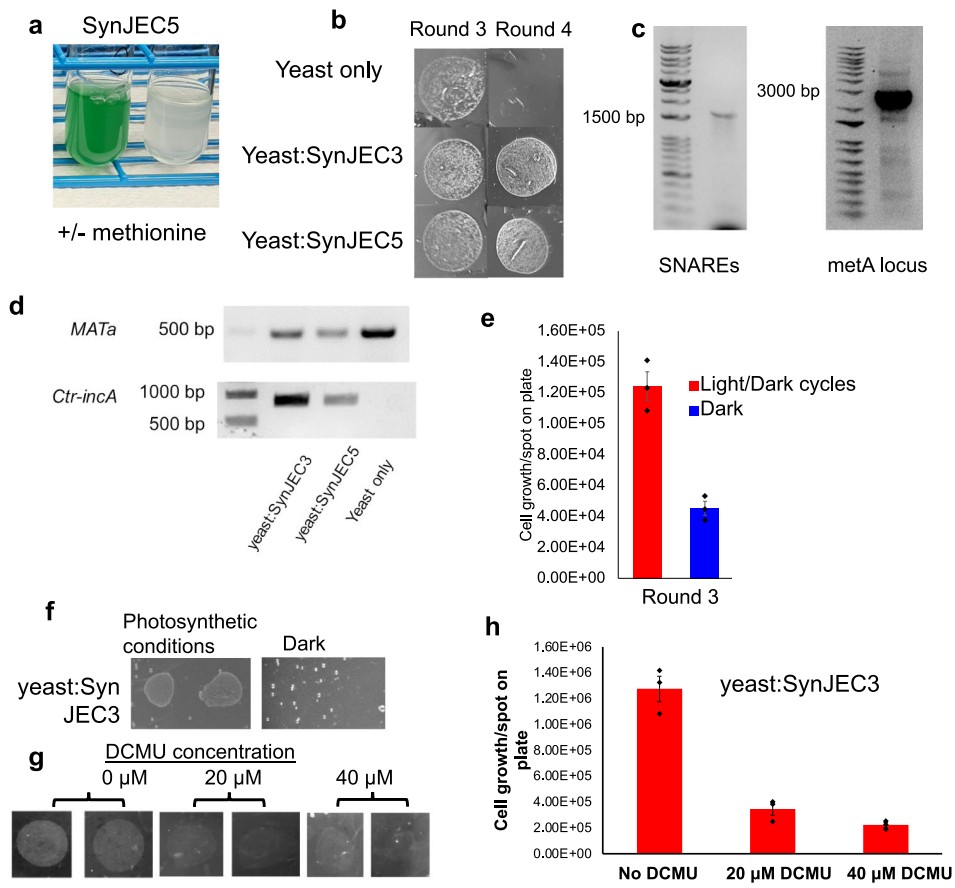

**Fig. 5 Rescue of respiration-deficient phenotype by yeast-*SynJEC* chimeras containing auxotrophic *SynJEC5* cells. a** Growth of *SynJEC5* cells in BG-11 medium in the presence and absence of L-methionine **b** Growth of yeast-*SynJEC3* and yeast-*SynJEC5* chimeras on Selection Medium III. The experiment was repeated three times independently with similar results. **c** Total DNA analysis of *SynJEC5* DNA reveals the presence of genomic SNARE-like proteins *Ctr-incA* and *CT813* (left) and disruption of the *metA* gene with a kanamycin resistance marker (right). The experiment was repeated twice independently with similar results. **d** Total DNA of yeast-SynJEC chimeras contains yeast *MATa* and *SynJEC Ctr-incA* genes. The experiment was repeated three times independently with similar results. **e** *S. cerevisiae cox2-60*-SynJEC3 chimeras are generated under and selected under photosynthetic conditions. After two rounds of growth *S. cerevisiae cox2-60*-SynJEC3 chimeras are propagated either under standard photosynthetic conditions or under dark; initial number of cells plated per spot is $10^4$ ($N = 3$ technical replicates; data are presented as mean values$+/-$ SEM). *P*-values were calculated by two-tailed *t*-test comparing the two means (95% CI, Cohen's $d = 6.2$, $P = 0.0016$). **f** SynJEC3 are starved for light for 72 h and the fused to *S. cerevisiae cox2-60* and selected under dark conditions (dark fusions). The *S. cerevisiae cox2-60*-SynJEC3 chimeras generated from dark fusions fail to grow whereas the control *S. cerevisiae cox2-60*-SynJEC3 chimeras propagated under photosynthetic conditions propagate as previously observed. The experiment was repeated twice independently with similar results. **g** Plate images to demonstrate the effect of (3-(3,4-dichlorophenyl)−1,1-dimethylurea), DCMU, on *S. cerevisiae cox2-60*-SynJEC3 chimeras. The experiment was repeated twice independently with similar results. **h** Cell count analysis to demonstrate the effect of (3-(3,4-dichlorophenyl)−1,1-dimethylurea), DCMU, on *S. cerevisiae cox2-60*-SynJEC3 chimeras ($N = 3$ technical replicates; data are presented as mean values$+/-$ SEM). *P*-values were calculated by two-tailed *t*-test comparing the two means. Source data are provided as a Source Data file.

significant insights into the genome evolution of the chromatophore of *Paulinella*[53,54]. For modern-day chloroplasts, studies suggest that *Gloeomargarita* are the closest identified relatives of modern-day chloroplasts[15,16]. While strains of *Gloeomargarita* and *Paulinella* are not readily genetically tractable, genetic tools have been developed to manipulate several strains of *Synechococcus*[17,18], which are relatives of *Gloeomargarita*[51,52]. These observations were central to our choice of engineering Syn7942 endosymbionts as a step towards investigating the endosymbiotic transformation of cyanobacteria into modern-day chloroplasts. As is evidenced from our observations described in this manuscript, this choice turned out to be an important consideration due to several innate abilities of Syn7942 that facilitated endosymbiosis. Conversely, when it came to deciding an appropriate host strain for endosymbiosis, our choice was determined mostly by genetic tractability, because it has been suggested that photosynthetic endosymbionts were established in

a variety of eukaryotic host cells during the chloroplast evolution[4,20]. Therefore, we decided to simply use a model eukaryotic cell, *S. cerevisiae*.

Though the modern day functions of chloroplasts are predominantly ATP synthesis, carbon assimilation, nitrate assimilation, sulfate assimilation, amino acid biosynthesis, fatty acid biosynthesis, it is widely suggested that bioenergetic considerations may have been the key drivers of organelle evolution[21,22]. Due to this wide range of chloroplast functions, it is still unclear what the exact drivers of endosymbiosis were. Through our investigations, we were able to engineer a yeast/cyanobacteria endosymbiotic platform where endosymbiotic cyanobacteria perform chloroplast-like functions for the host cells. Particularly, we demonstrated that SynJEC0 mutants lacking expression of any recombinant ADP/ATP translocase were able to transiently support the phenotypic rescue of *S. cerevisiae cox2-60* strains that were dependent on cyanobacterial endosymbionts for ATP. This

set of experiments suggested that cyanobacterial relatives of chloroplast precursors, already had an innate ability to establish bioenergetically relevant endosymbiosis with yeast mutants underdefined selection conditions. This hypothesis is supported by the presence of ATP/ADP translocase-like proteins or NTTs in Syn7942. Based on bioinformatics studies, proteins with homology to NTTs have been annotated as PBS lyase proteins in the cyanobacterial genome[24]. Our yeast/cyanobacteria endosymbiosis platform will facilitate systematic analysis of the importance of such cyanobacterial proteins to endosymbiosis. Although cyanobacteria are predicted to contain nucleotide transport fusion proteins (NTT, similar to ADP/ATP translocases), studies suggest that chloroplast NTTs were possibly derived from intracellular organisms like chlamydial[23,24]. We investigated this possibility by generating SynJEC1 strains that expressed recombinant *ntt1*, ADP/ATP translocase gene from an intracellular organism, *Protochlamydia amoebophila* UWE25, and observed that SynJEC1 endosymbionts were able to phenotypically rescue the growth of *S. cerevisiae cox2-60* strains that were respiration-deficient, and the chimeras had faster apparent growth rate and better stability. It is also suggested that precursors of organelles could have been parasitic in nature[55,56], and they may have genetic factors that allow the endosymbiont to establish a replicative niche within the host. To investigate this hypothesis, we generated a series of cyanobacterial mutants (SynJEC2, SynJEC3, SynJEC4 and SynJEC5) that recombinantly expressed various SNARE-like proteins from pathogenic intracellular bacteria along with ADP/ATP translocases and identified optimal cyanobacterial mutants that resulted in fastest apparent growth rate and stability. Under optimal conditions, the *S. cerevisiae*/Syn7942 chimera propagate through at least 15 to 20 generations of growth under defined photosynthetic growth conditions. Though the SNARE-like proteins are not necessary for yeast/cyanobacteria chimeras, the exact mechanism of how the expression of SNARE-like proteins provides improved stability of synthetic endosymbiosis is still unclear and is an active area of investigation[13,14]. It is possible that synthetic endosymbionts expressing SNAREs utilize alternate mechanisms that pathogens which result in enhanced stability of the host/endosymbiont chimera. We were also able to characterize the *S. cerevisiae*/Syn7942 chimeras through total internal reflection fluorescence microscopy, fluorescence confocal microscopy and TEM. Our data suggested that the cyanobacteria were structurally intact and did not appear to be entrapped in any intracellular organelles within yeast cells. We also developed a platform that created a metabolite interdependency between the host and the cyanobacterial endosymbiont, where the engineered cyanobacterial endosymbiont provided ATP to the host yeast cells and the yeast cells provided methionine to the cyanobacterial endosymbiont. Such platforms will enable metabolic coupling of the endosymbiont to the host, in a manner similar to metabolic coupling between the chloroplast and its photosynthetic eukaryotic cells. Our subsequent studies will also use similar approaches to recapitulate genome minimization events that could have occurred during chloroplast evolution.

Our efforts to engineer genetically tractable, artificial photosynthetic endosymbiotic systems could provide a platform to recapitulate various evolutionary trajectories related to the conversion of photosynthetic endosymbionts into photosynthetic organelles (i.e., chloroplasts). This endosymbiotic platform also has the potential to be further metabolically manipulated, analytically studied and imaged, and computationally modeled and predicted. Therefore, these platforms could facilitate laboratory evolutionary experiment to facility the bottom-up transformation of cyanobacterial endosymbionts into chloroplast-like organelles. For example, this platform could potentially be repurposed to study: (i) minimization of the endosymbiont genome, (ii) transfer of genes from the endosymbiont genome to the host genome, (iii) developing strategies to facilitate protein exchange between the endosymbiont and host, (iv) mutation-based evolution and selection to identify and characterize metabolic adaptations that could have occurred during organelle evolution, (v) the role of light in establishing photosynthetic endosymbionts and (vi) investigating if multiple endosymbionts can be established in host yeast cells to mimic secondary endosymbiosis[20] that is proposed to have occurred during chloroplast evolution, amongst others. Such efforts to recreate organelle evolution in lab could likely provide molecular-level understanding of how bacterial endosymbionts were transformed into chloroplasts, thereby resulting in the origin of photosynthetic eukaryotic life. Here it is important to note that there is no robust evidence to suggest that ATP synthesis was key driver of chloroplast evolution; in fact, it is still unclear what the exact drives of chloroplast were. There is robust sequencing evidence which suggests that eukaryotic host already had mitochondria to provide ATP when cyanobacterial endosymbionts were established. However, it is possible that cyanobacteria could have provided added advantage due to its light driven synthesis of ATP (photophosphorylation) unlike the mitochondria which performs oxidative phosphorylation. As precedented by the present-day synergistic interactions between chloroplast and mitochondria[57], it is even possible that cyanobacterial photosynthesis derived ATP and oxygen synthesis could have complemented mitochondrial oxidative phosphorylation process that synthesizes ATP. In addition to this, it is still unclear how exactly the endosymbiotic bacteria were internalized and stabilized within the host cells. Our current approach does not investigate the mechanisms that resulted in the internalization of cyanobacterial endosymbionts into eukaryotic cells and the establishment of these endosymbionts in the cytoplasm.

In addition to the evolutionary insights, this genetically tractable photosynthetic endosymbiotic platform can be repurposed for various synthetic biology applications as well; for example, such artificial, genetically tractable photosynthetic platforms can be used for biosynthesis and bioproduction applications. These studies could also provide a genetically tractable experimental platform for synthetic biology efforts on designing minimal bacterial genomes to attain 'minimal organism' to understand minimum requirements necessary to support photosynthetic life in privileged environments of the host cytosol[58]. In summary, our studies will provide a roadmap to use first principles of endosymbiotic theory to convert non-photosynthetic organisms into new photosynthetic life-forms.

## Methods

**Strains**. *Synechococcus elongatus* strains were derived from *S. elongatus* PCC 7942 (Syn7942). This strain was obtained from Prof. Susan Golden's lab (University of California San Diego, UCSD). We used *S. cerevisiae* ρ + NB97 (*MATa leu2-3,112 lys2 ura3-52 his3ΔHindIII arg8Δ::URA3 [cox2-60]*) as a host for *S. elongatus* endosymbionts. The *S. cerevisiae-cox2-60* strain was obtained from the Schultz lab (Scripps Research).

**Growth media**. *S. elongatus* cells were grown in sterile Erlenmeyer flasks containing liquid BG-11 medium. These cultures were shaken aerobically at 37 °C and 250 rpm under 3000 lux. Yeast cells were shaken aerobically at 30 °C and 250 rpm in YPD medium (1% yeast extract, 2% peptone, 2% glucose) containing 50 mg/L carbenicillin.

List of all the fusion selection media:

Selection medium I: 1% yeast extract, 2 % peptone, 3 % glycerol, 0.1 % glucose, 1 M sorbitol, 2 % agar, and 1X BG-11 salts.

Selection medium II: 1% yeast extract, 2 % peptone, 3 % glycerol, 0.1 % glucose, 1 M sorbitol, 2 % agar, 1X BG-11 salts, and 50 mg/ml carbenicillin.

Selection medium III: 1% yeast extract, 2 % peptone, 3 % glycerol, 1 M sorbitol, 2 % agar, 1X BG-11 salts, and 50 mg/ml carbenicillin.

NSII-recombinant *S. elongatus* mutant cultures were supplemented with 7.5 mg/L chloramphenicol. Unless noted otherwise, kanamycin-resistant *S.*

*elongatus* mutant cultures were supplemented with 50 mg/L kanamycin and 2 mg/L L-methionine.

When noted, light-starved Syn7942 cells were inoculated (OD$_{730}$ = 0.2) in BG-11 medium, wrapped in aluminum foil and incubated for 4 d, without shaking, at 30 °C. When noted, yeast media was supplemented with adenine sulfate (Alfa Aesar A16964-09, 20 mg/L), Antimycin A (Sigma A8674-25MG), or DCMU (A2B Chem AG00409). Yeast cells transformed with pML64 were selected on synthetic defined (SD) medium containing 0.67% yeast nitrogen base without amino acids (Sigma Y0626-250G), L-lysine (Sigma L5501-5G, 60 mg/L), L-histidine (Sigma H8000-10G, 20 mg/L), L-arginine (Sigma A-3704, 20 mg/L), uracil (Sigma U1128-25G) and 3% glycerol.

**Construction of plasmids.** Double-stranded and single-stranded DNA oligonucleotide fragments were purchased from Integrated DNA Technologies (IDT). Defined DNA fragments were amplified using PCR (Q5 Hot Start High-Fidelity 2X Master mix, NEB catalog # M0494S) and inserted into defined sites in the host vectors using Gibson assembly[59]. Double-stranded oligonucleotide sequences (gblocks) used for Gibson assembly are listed in Supplementary Data 1. Genomic DNA fragments used in cloning are listed in Supplementary Table 1. Cyanovectors were obtained from Prof. Susan Golden's lab (UCSD). Where noted, coding sequences were codon-optimized for *S. elongatus* expression using IDT codon optimization software (https://www.idtdna.com/CodonOpt). All vectors were transformed into One Shot® *ccdB* Survival™ 2 T1$^R$ Chemically Competent Cells (Invitrogen A10460) according to manufacturer's specifications. The oligonucleotides used in plasmid construction are listed in Supplementary Data 2. The plasmids pML3-pML28 are derived from the Cyanovector pCV0055, pML58 and pML62 are derived from the CyanoVector pCV0049. Vector maps are included in Supplementary Fig. 1, and detailed vector map links are provided in Supplementary Table 3.

pML3: pCV0055[18] was linearized by PCR using the oligonucleotides AM1216/AM1217. A gBlock of the *UWE25-ntt1* gene codon optimized for *S. elongatus* was amplified by PCR using the oligonucleotides AM1214/AM1215. The amplified DNA fragment was inserted into linearized pCV0055 by Gibson assembly to afford pML3.

pML14: pML3 was linearized by PCR using the oligonucleotides AM1195/AM1312. A gBlock of the *C. trachomatis incA* gene codon-optimized for *S. elongatus* (*Ctr-incA*) was amplified by PCR using the oligonucleotides AM1313/AM1314 and inserted into linearized pML3 by Gibson assembly to afford pML14.

pML17: pML14 was linearized by PCR using the oligonucleotides AM1312/AM1351. A gBlock of the *CT_813_CDO-1* gene was amplified by PCR using the oligonucleotides AM1352/AM1353 and inserted into linearized pML14 by Gibson assembly to afford pML17.

pML28: pML17 was linearized by PCR using the oligonucleotides AM1506/AM1507. A variant of the *C. caviae incA* gene codon-optimized for *S. elongatus* (*Cca-incA*) was amplified by PCR using the oligonucleotides AM1502/AM1503, purified and amplified once more using the oligonucleotides AM1504/AM1505. The amplicon was inserted into linearized pML17 by Gibson assembly to afford pML28.

pML62: pCV0049 was linearized by PCR using the oligonucleotides JC205/JC206. A 484-bp fragment of the 5′-end of the Syn7942 *metA* gene (*Synpcc7942_0370*) was amplified from purified Syn7942 genomic DNA using the oligonucleotides JC204/JC207. The amplicon was inserted into linearized pCV0049 by Gibson assembly and the construct was subsequently linearized using the oligonucleotides JC200/JC203. A 530-bp fragment of the 3′-end of *Syncpcc7942_0370* was amplified from genomic DNA using the oligonucleotides JC201/JC202 and the amplicon was inserted into the linearized vector by Gibson assembly to afford pML62.

pML58: pML62 was linearized by PCR using the oligonucleotides AM1531/AM1532. The PCR product was purified and treated with T4 Polynucleotide Kinase (NEB #M0201S), T4 Ligase (NEB #MO202S) and DpnI (NEB #R0176S) for 1 h at room temperature to produce the ligated plasmid. The plasmid was then linearized by PCR using the oligonucleotides JC151/JC152. An origin of transfer (oriT) domain was amplified by PCR from the commercial plasmid pSET152 using the oligonucleotides JC153/JC154. The amplicon was inserted into the linearized plasmid by Gibson assembly to afford pML58.

pML63 is derived from the commercial plasmid pRS425, which was donated by the van der Donk lab (UIUC). The 2×Su9-MTS-LP-*COX2*-W56R expression cassette was first reported by Supekova and Schultz[35].

pML59: pML14 was linearized by PCR using the oligonucleotides JC221/JC233. The linear PCR product was purified, amplified with JC221/JC226 and ligated by KLD reaction to afford the FLAG-tagged *Ctr-incA* construct pML59.

pML60: pML17 was linearized by PCR using the oligonucleotides AM1312/JC235 and ligated by KLD reaction. The construct was then linearized by PCR using the oligonucleotides JC223/JC233. The linear PCR product was purified, amplified with JC223/JC226 and ligated by KLD reaction to afford the FLAG-tagged *CT_813* construct pML60.

pML63: pRS425 was linearized by PCR using the oligonucleotides JC249/JC248. The *S. cerevisiae ACT1* promoter was amplified from purified *S. cerevisiae* genomic DNA by PCR using the oligonucleotides JC246/JC247. The amplified DNA fragment was inserted into linearized pRS425 by Gibson assembly and the construct was subsequently amplified by PCR using the oligonucleotides JC250/JC251. A gBlock of 2×Su9-MTS-LP-*COX2*-W56R was amplified by PCR using the oligonucleotides JC254/JC255 and the amplified DNA fragment was inserted into the linearized plasmid by Gibson assembly to afford pML63.

pML64: pML63 was amplified by PCR using the oligonucleotides JC128/JC273. The *S. cerevisiae TPI1* promoter was amplified from purified *S. cerevisiae* genomic DNA by PCR using the oligonucleotides JC272/JC127. The amplified DNA fragment was inserted into linearized pML63 by Gibson assembly to afford pML64.

**Site-directed mutagenesis in cyanobacteria.** Chromosomal integration of genes in Syn7942 was achieved using a modification of the method of Golden[60]. Briefly, a portion of Syn7942 overnight culture (15 mL) was centrifuged for 10 min at 3000 × *g* and 24 °C. The pellet was washed once with 10 mL NaCl (10 mM) and then resuspended in 0.3 mL BG-11 at room temperature. To this suspension was added purified plasmid (1.5 µL) containing genes flanked by NSII recombination sites. The mixture was added to a 1.5 mL microcentrifuge tube and shaken in the dark (12–16 h at 70 rpm). Cells transformed with NSII plasmids were added to glass test tubes containing liquid BG-11 (5 mL, room temperature) and antibiotic and cultured under normal conditions for 4 d, at which point surviving cells were added to Erlenmeyer flasks and grown under normal conditions. Cells transformed with pML58 were not viable when rescued in liquid BG-11 medium; instead, the cell-plasmid mixture was spread on BG-11-agar medium supplemented with kanamycin and L-methionine and incubated at 30 °C under 3000 lux for ≤10 days. The quality of *S. elongatus* cultures was evaluated regularly by microscopy, streaking the cells onto BG-11-agar and by PCR analysis of recombinant loci (oligonucleotides used for PCR analysis are listed in Supplementary Table 4).

SynJEC0 was generated by transformation of wild-type Syn7942 with pCV0055 to give a chloramphenicol-resistant recombinant mutant.

SynJEC1 was generated by transformation of wild-type Syn7942 with pML3 to give a recombinant mutant which ectopically expresses an ADP/ATP translocase from the NSII locus.

SynJEC2 was generated by transformation of wild-type Syn7942 with pML14 to give a recombinant mutant which ectopically expresses *Ctr-incA* and an ADP/ATP translocase from the NSII locus.

SynJEC3 was generated by transformation of wild-type Syn7942 with pML17 to give a recombinant mutant which ectopically expresses *Ctr-incA*, *CT813* and an ADP/ATP translocase from the NSII locus.

SynJEC4 was generated by transformation of wild-type Syn7942 with pML28 to give a recombinant mutant which ectopically expresses *Ctr-incA*, *CT813*, *Cca-incA* and an ADP/ATP translocase from the NSII locus.

SynJEC5 was generated by transformation of SynJEC3 cells with pML58. The cells were plated on BG-11-agar medium supplemented with chloramphenicol (5 mg/L), kanamycin (5 mg/L) and L-methionine (2 mg/L), with individual colonies appearing within 10 days. Colonies were extracted and spotted on fresh BG-11 agar supplemented with the same components. After 7 days, the resulting dark green spots were added to liquid BG-11 (1 mL) and incubated under normal culturing conditions for Syn7942. The cells were passaged every 4 days into fresh BG-11 medium (5 mL) with increasing concentrations of kanamycin (5, 25 and finally 50 mg/L) in order to eliminate all chromosomal copies with intact *Synpcc7942_0370*. During each passage, the *Synpcc7942_0370* locus was amplified by PCR using the oligonucleotides LL56/LL57. SynJEC5 cultures were considered 'pure' after 5 rounds of passaging, at which point only recombinant amplicons were produced by PCR. Methionine auxotrophy of SynJEC5 was evaluated by washing the cells in BG-11 medium and inoculating cultures containing chloramphenicol and kanamycin in the presence and absence of L-methionine (Supplementary Fig. 5a).

SynJEC9 was generated in a similar manner as SynJEC5 by transformation of SynJEC1 cells with pML58 instead of pML62. As with SynJEC5, the cells were plated on BG-11-agar medium supplemented with chloramphenicol (5 mg/L), kanamycin (5 mg/L) and L-methionine (2 mg/L). Individual colonies appeared within five days, which were then extracted and spotted on fresh BG-11 agar supplemented with kanamycin (5 mg/L) and L-methionine (2 mg/L). Afterwards, the cells were passaged through multiple rounds of growth in liquid BG-11 medium in the same manner as SynJEC5 until PCR analysis of genomic DNA with the oligonucleotides LL56/LL57 produced only a recombinant band at the *Syncpcc7942_0370* locus (Supplementary Fig. 5a).

**Measurement of ATP release by cyanobacteria mutants expressing ATP/ADP translocase after ADP challenge.** In order to eliminate contaminant ATP, ADP solution (Sigma A2754) was treated with hexokinase according to the following protocol[61]. ADP (80 mM, pH 7.5) was incubated with D-glucose (200 mM), MgCl$_2$ (2 mM) and hexokinase (Sigma H4502-500UN) (0.04 U/µL) at room temperature for 2 h. The solution was then filtered through an Amicon Ultra 0.5 column (14,000 × *g*, 15 min) to eliminate hexokinase.

SynJEC0 and SynJEC1 cells were grown for 3 d to reach densities of ~30,000,000 cells/mL. For each assay, 300,000,000 cells were harvested by centrifugation (3000 × *g*, 5 min, room temperature), the supernatant was aspirated and the pellet was washed once with 20 mM Tris-HCl. The cells were then incubated with ATP solution (Sigma G8877) (10 mM, pH 7.5) for 30 min at 37 °C, washed three times with 20 mM Tris-HCl to eliminate extracellular ATP. ADP (80 µM final

concentration) was added and the cells were incubated statically at 37 °C. The mixtures were then centrifuged (10,000 × g, 5 min) and the supernatant ATP concentration was determined by luciferase assay (ATP determination kit, Life Technologies - #A22066). ATP standards (provided with the kit) were used obtain a calibration curve.

**Introduction of mutant cyanobacteria to *S. cerevisiae* cells.** We adapted a method for generating yeast-*E. coli* chimeras to be used with *S. elongatus*[13,14]. SynJEC0-9 mutants were grown under constant light, with shaking, for 4 d. After this time, the cells (30 mL) were harvested (3000 × g, 10 min, 24 °C), washed twice with BG-11 and resuspended in BG-11 (500 μL). *S. cerevisiae cox2-60* cells were grown aerobically in YPD medium (120 mL) for 24 h. The yeast was harvested (4696 × g, 10 min, 24 °C), washed twice with sterile water, twice more with SCEM (1 M sorbitol, 13 mM β-mercaptoethanol) and resuspended in ice-cold sterile-filtered SCEM solution (10 mL) containing Zymolyase 100 T (15 mg per gram of yeast pellet). The suspension was incubated for 1 h at 37 °C to give spheroplasts. The suspension was then cooled on ice for 30 min and centrifuged for 10 min at 1500 × g and 4 °C. The pellet was washed twice, gently, with chilled SCEM and resuspended in SCEM (2.5 mL) The suspension would remain usable for fusions for at least 24 h if kept refrigerated.

The spheroplast suspension (750 μL) was mixed with chilled TSC buffer (10 mM Tris-HCl, 10 mM CaCl₂, 1 M sorbitol, pH 8) (750 μL) and incubated for 10 min at 30 °C. This mixture was then centrifuged (1500 × g, 10 min) and the supernatant was carefully discarded. The spheroplasts were resuspended in room-temperature TSC buffer (120 μL) and sorbitol (4 M, 60 μL). Dense *S. elongatus* cell suspension (120 μL) was added quickly to the spheroplast suspension, mixed by tube inversion and incubated for 30 min at 30 °C. This mixture was then decanted into a round-bottom polypropylene tube containing PEG buffer (20% PEG 8000, 10 mM Tris-HCl, 2.5 mM MgCl₂, 10 mM CaCl₂, pH 8) (2 mL) and incubated without shaking at 30 °C for 45 min. The cells were centrifuged (1500 × g, 10 min, 24 °C), the supernatant was discarded and YPDS (YPD with 1 M sorbitol added) was added gently without disrupting the pellet. The cells were incubated under light without shaking for 2 h at 30 °C. After this time, the pellet was partially dislodged by flicking the tube. The mixture continued to incubate for 3 h with shaking (70 rpm), after which time the cells were harvested (1500 × g, 10 min, 24 °C), resuspended in 1 M sorbitol (300 μL) and plated on Selection-I medium. After drying for 5 min, a second layer of Selection-I medium was overlaid on top of the cells. The plates were incubated at 30 °C in a 12 h light-dark cycle for at least four days, until colonies appeared between the agar layers. The colonies were extracted from the agar, suspended in 1 M sorbitol and spotted on Selection-II medium. For subsequent rounds of propagation, cells were scraped from the surface of the agar, resuspended in 1 M sorbitol and spotted on Selection-III medium.

**Cell count of *S. cerevisiae cox2-60*/cyanobacteria chimeras.** Spots on Selection-III medium were removed manually from the agar plate and placed intact inside a 1.5-mL microcentrifuge tube. Sorbitol (1 M, 200 μL) was added to the surface of the agar and cells were removed by pipetting. The tube was briefly centrifuged (5 s) to remove the cell-containing flowthrough from the agar. Cells were then mounted on a reusable glass slide and counted in triplicate from a brightfield image using the Countess II FL Automated Cell Counter (Fisher cat. # AMQAF1000) per the manufacturer's instructions.

**Total genomic DNA isolation and PCR analysis.** Total DNA isolation of chimeras was achieved using the Yeast DNA Extraction Kit (Thermo Fisher 78870) using the manufacturer's protocol. Cells were scraped from the surface of the agar, harvested (5000 × g, 5 min), resuspended and incubated (65 °C, 10 min) with Y-PER Reagent (8 μL/mg pellet). The mixture was centrifuged (13,000 × g, 5 minutes), the supernatant was discarded and the pellet was resuspended and incubated (65 °C, 10 min) with DNA Releasing Reagent A (16 μL) and DNA Releasing Reagent B (16 μL). Protein Removal Reagent (8 μL) was added to the mixture which was then centrifuged (13,000 × g, 5 min). The supernatant was then added to a clean microcentrifuge tube and DNA was precipitated by adding iso-propyl alcohol (24 μL). The mixture was centrifuged (13,000 × g, 10 min) and the pellet was washed by adding 70% ethanol (13,000 × g, 1 min). The supernatant was then removed by aspiration and the pellet was allied to dry inside a fume hood. TE buffer (10 mM Tris, 1 mM EDTA, 50 μL) was added to the pellet to afford the total isolated DNA. PCR analysis was performed using oligonucleotides listed in Supplementary Table 2.

**Growth curves for yeast and cyanobacteria cells.** Growth curve data were obtained by growing 200 uL of cells in sterile 96-well plates (Nunclon 167008). Yeast cells were grown aerobically with shaking at 30 °C and 240 rpm. Syn7942 cells were shaken at 30 °C and 70 rpm. Measurements were all obtained with the BioTek Synergy H1M Microplate Reader and Gen5 v.3.11.19 software. Pathlength correction for these measurements was determined manually. Water (200 μL) was added to a well and A975 and A900 were measured by plate reader. Water was also added to a quartz cuvette (Spectrocell R-3010-T) and the near-IR absorption spectrum was obtained using the Shimadzu UV-VIS-NIR Spectrophotometer

UV-3600 and UVProbe v.2.34 software. Corrections to absorbance data were made with the following equation:

$$A_{correct} = A_{raw} * \frac{A_{975}(cuvette) - A_{900}(cuvette)}{A_{975}(well) - A_{900}(well)} \quad (1)$$

**Analysis of yeast/cyanobacteria chimeras using pTIRF microscopy.** Samples were prepared by extracting the chimeric cells from plates and washing them once with 1 M sorbitol. They were analyzed with a home-built TIRF microscope based on a Zeiss Axiovert 200 M stand. A Cobolt diode-pumped 561 nm laser was used in this work. The laser beam is combined with other two laser lines in the system by using a set of Semrock LaserMUX™ filters, and then sent through an acousto optic tunable filter (AOTF, Quanta Tech Inc). The AOTF is used through the microscope control software to simultaneously and independently control the power of each laser on sample. After the AOTF, a laser speckle-reducer (LSR-3005-17S-VIS, Optotune) is used with a set of achromatic lenses to provide homogeneous illumination across the entire field of view for TIRF or pTIRF microscopy. The dichroic beamsplitter used in the system is a Semrock LF405/488/561/635-B-000 and the emission filter is a Chroma bandpass filter HQ 653/95 nm. The pTIRF images were acquired with a Photometric 512 Evolve EMCCD camera. Samples were viewed and imaged using a 100X oil immersion objective lens with NA = 1.4. The software for microscope control and data acquisition was developed using C++ and Labview. All images were processed with Fiji, for example, to merge a qTIRF image and a brightfield image that were acquired from the same sample position. Fluorescence of chimeras was measured using ImageJ 1.53c. Images of chimeras excited with either 405 nm or 561 nm laser were concatenated and fluorescent regions of interest (ROIs) were identified by thresholding.

**Analysis of yeast/cyanobacteria chimeras using fluorescence confocal microscopy.** Stock solutions (40X) of Concanavalin A, Fluorescein Conjugate (ConA; Thermo Fisher C827) were prepared by dissolving the lyophilized powder in sodium bicarbonate (0.1 M) to a final concentration of 2 mg/mL. These solutions were stored at −20 °C without exposure to light. Directly prior to use in sample preparation, the stock solution was thawed and centrifuged for 10 s. Microscopy samples were prepared by extracting the chimeric cells from plates and washing them once with Hank's Buffered Salt Solution (HBSS; NaCl (140 mM), KCl (5 mM), CaCl₂ (1 mM), MgSO₄ heptahydrate (0.4 mM), MgCl₂ hexahydrate (0.5 mM), Na₂PO₄ dihydrate (0.3 mM), KH₂PO₄ (0.4 mM), D-glucose (6 mM), NaHCO₃ (4 mM)). The cells were then incubated (37 °C, 10 min) in HBSS supplemented with ConA to a final concentration of 50 μg/mL, washed twice with HBSS, centrifuged and incubated with Karnovsky fixative (2% glutaraldehyde, 2.5% paraformaldehyde). Samples were analyzed with a commercial Leica SP8 fluorescence confocal microscope. Samples were viewed and imaged through a 63X/1.40 HC PL APO Oil CS2 lens and excited with 488 nm and 561 nm laser light. Emission wavelengths in the 510/20 nm range were detected with photomultiplier tube (PMT) detector, and emission wavelengths in the 650/20 range were detected using a high-sensitivity GaAsP HyD detector. Leica Application Suite (LASX) was used to collect raw data. All images were processed with Fiji to display an overlay of the two channels.

**Analysis of yeast/cyanobacteria chimeras using TEM.** Cells were extracted from agar plates in HBSS (700 μL) and pelleted by centrifugation (6000 × g, 5 min). The supernatant was removed and the cells were resuspended in Karnovsky fixative (20 μL) and incubated for 30 min. The cells were then pelleted and resuspended in HBSS. The medium was removed and replaced with a fixative (2.5% EM-grade glutaraldehyde and 2.0% EM-grade formaldehyde in 0.1 M sodium cacodylate buffer, pH 7.4) for 3 h at 4 °C. The fixative was then removed, replaced briefly with buffer, and then replaced with 1% osmium tetroxide in buffer for 90 min. Each sample was then subjected to 10-minute buffer rinse, after which it was placed in 1% aqueous uranyl acetate and left overnight. The next day, each sample was dehydrated via a graded ethanol series, culminating in propylene oxide. Following a graded propylene oxide; Epon812 series, the nuclear pellets were embedded in Epon812 prior to cutting. Ultrathin (ca. 90 nm) Epon sections on grids were stained with 1% aqueous uranyl acetate and lead citrate[62]. After the grids dried, areas of interest were imaged at 160 kV, spot 3 using a Philips/FEI (now Thermo Fisher FEI) Tecnai G2 F20 S-TWIN transmission electron microscope in the Microscopy Suite at the Beckman Institute of Advanced Science and Technology (University of Illinois at Urbana-Champaign).

**Growth dependence of chimeras on light.** For partial light starvation after the introduction on cyanobacteria, SynJEC3 cells were introduced to yeast spheroplasts in the manner described, plated on and overlaid with Selection-I medium. The plates were incubated at 30 °C for 4 d inside a box covered in aluminum foil placed into the incubator. After this time, the contents of the overlaid agar plate were extracted by stabbing and plated on Selection-II medium grown either in 12 h day/night cycle or under the same conditions described to remove light. The growth of chimeras under these different conditions was quantified in the manner described. For complete light starvation, a flask containing BG-11 medium was inoculated with log-phase SynJEC3 cells to OD₇₃₀ of 0.200 and incubated for 4 d without shaking inside a box covered in aluminum foil. These cells were introduced to yeast

spheroplasts in the manner described, plated on Selection-I medium and incubated for 4 d at 30 °C inside a foil-covered box. After this time, the contents of the plate were extracted by stabbing the agar and plated on Selection-II medium. The plates were incubated for 4 d more in the absence of light.

**Reporting summary**. Further information on research design is available in the Nature Research Reporting Summary linked to this article.

## Data availability

Data supporting the findings of this work are available within the paper and its Supplementary Information files. A reporting summary for this article is available as a Supplementary Information file. Source data are provided with this paper.

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

## Acknowledgements

This work was supported by the Moore–Simons Project on the Origin of the Eukaryotic Cell, GBMF9732, grant https://doi.org/10.37807/GBMF9732. This funding was received by A.P.M. The research reported in this publication was also supported by the National Institute of General Medical Sciences of the National Institutes of Health under Award Number R01GM139949. This funding was received by A.P.M. The content is solely the responsibility of the authors and does not necessarily represent the official views of the National Institutes of Health. A.P.M. thanks Prof. Peter Schultz (Scripps Research), Dr. Lubica Supekova (Scripps Research), Prof. Susan Golden (UCSD) and Prof. Wilfred van der Donk (UIUC) for comments and discussions.

## Author contributions

A.P.M., J.E.C., S.D.A. Y.G. designed experiments. J.E.C., S.D.A., Y.G., G.H.L, N.T.H. performed biochemical experiments. J.E.C., D.Z. performed pTIRF and confocal microscopy experiments and obtained data. J.E.C. prepared samples for TEM experiments and C.L.W performed TEM experiment and obtained and analyzed TEM data. A.P.M., J.E.C., S.D.A., C.L.W., D.Z. analyzed the data and wrote the manuscript.

## Competing interests

Authors declare no competing interests.
