## [Peer Review File · Nature Communications]

Engineering artificial photosynthetic life-forms through endosymbiosisEditorial Note: This manuscript has been previously reviewed at another journal that is not operating a transparent peer review scheme. This document only contains reviewer comments and rebuttal letters for versions considered at Nature Communications.

Reviewers' Comments:

Reviewer #1:

Remarks to the Author:

The authors have answered some of my previous comments; however, I still have concern over this manuscript.

I previously asked for the rationale for targeting ATP translocation, as this is not what is done in nature via chloroplasts and their host cells. Although the authors cite Refs 22 and 23 to establish that there are "bioenergetic considerations" in organelle evolution, this seems a bit vague. It's largely established that the main advantage of having a photosynthetic endosymbiont/organelle is photosynthesis for C assimilation. The authors have not provided an example where ATP is translocated to establish an endosymbiosis. The reason this is important, is because the entire introduction is written as a means of recapitulating the endosymbiosis process; however, the authors don't provide sound rationale or direct precedence for assuming ATP translocation would be the mechanism for this unique evolutionary process. If this manuscript were written purely as an engineering paper, then I would not be as worried; however, this is not the case. From an engineering perspective, the results are interesting, as providing a new way to increase ATP availability may have many applications. Unfortunately, the paper (specifically the Introduction) is currently written to have large evolutionary implications, which don't seem well supported.

I believe much of the confusion stems from the authors not directly stating that they are trying to complement mitochondrial endosymbiosis (which drives ATP translocation in eukaryotes), not plastid endosymbiosis (which drives C translocation). The authors need to fundamentally rewrite how they present the paper in the Introduction to clarify these points (rather than to just input a few sentences) to not mislead readers. There is a disconnect between the motivations set in the Introduction and the Experiments described. Specifically, increasing ATP translocation from a cyanobacterial endosymbiont is not teaching us about plastid endosymbiosis, because the original eukaryotic host already had a mitochondrion to provide ATP. Thus, I believe it is more appropriate to frame this study as an engineering study to see the implications of increasing ATP accumulation in the host when driven by photosynthesis.

One reason there might be a limited number of doublings that ADP and Pi is running out in the cyanobacteria. Can the authors show this? If so, then the authors will have provided sufficient proof of the mechanism being ATP translocation. Is there a way to add excess ADP/Pi to the yeast to see if this supports further generations in the chimera?

The DCMU studies are helpful in establishing the connection that the described phenomenon is photosynthetically driven. This is perhaps the single most convincing data of the manuscript. However, one question that the authors did not address in my first-round of reviewer comments was why the doublings in the are measured by cell count/spot on plates. It seems from Fig S10, it was possible to do liquid growth curves, so it's not clear why there isn't consistency in how growth was measured.

The authors say that they have added text describing the loss of endosymbionts, but all they have written is: "As per our genomic DNA analysis of the chimeras, after more than 20 rounds of doublings, we observe that the chimeras lose their viability possibly due to loss of cyanobacteria." This does not

address my original concern that the authors have not looked at what is happening to the endosymbiont; rather, they are only speculating that there is a loss of cyanobacteria. Currently, the manuscript is still superficially descriptive. Although mechanistic understanding of what the observed transient growth in chimeras is not required for this to be a paper of interest, more thorough understanding of what is happening in this system is required for it to be more scientifically rigorous. The authors say that they are setting up a microscopy-based system to track live cells; however, this is beyond the scope of the manuscript. I agree this would be beyond the scope of the paper; however, my original comment was not to do single cell live tracking. I think providing wide-field micrographs at different time points to observe how the endosymbiont supposedly is lost over 20 doublings would suffice. This seems doable seeing that TIRF and confocal images could be collected for Figure 4.

I believe that the claims of this manuscript are quite bold. The addition of the DCMU experiment assuages some of my concerns, but even just adding observational data of the endosymbiont of 20 doublings would provide much needed evidence that the system is functioning the way they describe it.

Reviewer #3:

Remarks to the Author:

Thank you for adding the TEM, I think this is important.

But, as expected, since the cyanobacteria are located in the cytoplasm, and since they have completely skipped phagocytosis, the discussion of SNAREs in the manuscript does not make sense. It is true, as the authors say, that "Literature suggests that one mechanism by which intracellular pathogenic bacteria avoid lysosomal degradation is through the expression of SNARE-like proteins, which are thought to inhibit SNARE-mediated membrane fusion through mimicry of the host SNAREs." But, as described in reference 38 cited by the authors, "Intracellular pathogens enter cells via phagocytosis. In their host cells, a significant number of intracellular microorganisms such as Mycobacterium, Salmonella, Chlamydia or Legionella, have the ability to create favorable intracellular compartments within which to multiply. This strategy relies on the capacity of bacteria to block undesirable fusion events with the cell's degradative compartments and their capacity to create new advantageous fusion pathways with host vesicles."

Since these cyanobacteria have not entered the cell through phagocytosis, they have not had the opportunity to remodel their vacuolar compartment, and so while expression of SNAREs might be helping the stability of this system, it is quite unclear why it is helping, and it is likely that the reason is completely unrelated to the reasons SNAREs are implicated in pathogen survival.

Reviewer #4:

Remarks to the Author:

General comments:

In this study, Cournoyer and colleagues constructed a hybrid represented by yeast cells containing an introduced autotrophic cyanobacterium *Synechococcus* PCC 7942. The main issue raised by Rev. 2 was the lack of evidence for the photosynthetic activity of the inserted cyanobacterium which should be essential for the growth of the hybrid cells. The revised version of the manuscript now provides several lines of evidence for this activity, which is summarized in the new paragraph "Growth dependence of the yeast/cyanobacteria chimeras on photosynthesis" on page 8. (i) Authors assessed the growth of the hybrid under day/night cycle and in the dark. Fig. 5 shows that light significantly

accelerated growth of the organism. When the cyanobacterial cells were pre-adapted to the darkness before the fusion, the resulting hybrids did not grow in the dark at all. This can be considered rather convincing evidence for the photosynthetic activity of the cyanobacterium as the crucial factor for the hybrid survival. (ii) Addition of inhibitor of photosystem II DCMU into the growth medium resulted in a significant retardation of the hybrid light-dependent growth. (iii) Authors monitored fluorescence of cyanobacterial phycobilisome antennas and chlorophyll embedded within the cyanobacterial photosystems. This indicates an efficient energy transfer between phycobilisomes and photosystems indicating photosynthetic competence of the internalized cyanobacteria. Nevertheless, in the last remark of Rev. 2 time dependent fluorescence based assays at a single cell level were recommended as an excellent tool for evaluating the active photosynthesis (photosystem II). This would be, for instance, induction of fluorescence by illumination after previous dark adaptation, which unfortunately was not performed by the authors but would provide the convincing evidence for the photosynthetic activity in situ. Nevertheless, I consider the new experiments and their results sufficient for proving the need of cyanobacterial photosynthetic activity for growth of the yeast:Syn chimera, which was the main request of the Rev. 2.

An error for correction:

1. In the legend to Fig. S4 please correct:... fluorescence microcopy...

We thank the reviewers for taking the time to read our revised manuscript and providing critiques. Please see our responses below (in blue). In our responses, we have listed the changes we have made to our resubmitted manuscript.

REVIEWER COMMENTS

Reviewer #1 (Remarks to the Author):

The authors have answered some of my previous comments; however, I still have concern over this manuscript.

Based on Reviewer #1 comments during first round of review, we performed following experiments:

1. Added experimental data that demonstrates that reconstitution of ATP synthesis drives the phenotype of *cox2-60* strain (Fig. 2E).
2. We calculated exact doubling numbers (Table S7)
3. We calculated doubling times for engineered *cox2-60* strain containing pML64 which reconstitutes ATP synthesis (Fig. S8)
4. We calculated doubling times for engineered yeast/cyanobacteria chimeras (Fig. 2, 3 and listed in text)
5. To determine “intracellular positioning”, we performed TEM experiment (Fig. 4C)
6. We extensively characterized the dependence of chimeras on photosynthesis for propagation (Fig. 5E-G)
7. We characterized the importance of light (and hence photosynthesis) for the viability of chimeras on photosynthesis for propagation (Fig. 5E-G) using DCMU as a probe
8. To “track the fate of the cyanobacteria over time. Are the yeast dying?” – we answered this question by using PCR analysis of total genomic DNA. As per Reviewer #1 clarification, we have now added widefield images to track chimeras.

As far as the authors could tell, the only experiment that the authors did not perform was live cell tracking experiment (which we may have misunderstood as a request in the first place; we thank Reviewer #1 for the clarification, see below). As Reviewer #1 mentions below, and we agree with Reviewer #1, that this is beyond the scope of this paper.

In addition to the above experiments, we added citations and revised our introduction and discussion in resubmission to highlight the role of bioenergetics in chloroplast evolution and discuss the evolution of functional role of endosymbionts as they transformed into present-day organelles. For this resubmission, we have extensively modified our introduction and discussion to highlight engineering aspect of this study.

For resubmission 2, we have made changes to the text of the manuscript and added data to the SI to address Reviewer #1 comments (see below):

I previously asked for the rationale for targeting ATP translocation, as this is not what is done in nature via chloroplasts and their host cells. Although the authors cite Refs 22 and 23 to establish that there are “bioenergetic considerations” in organelle evolution, this seems a bit vague. It’s largely established that the main advantage of having a photosynthetic endosymbiont/organelle is photosynthesis for C assimilation. The authors have not provided an example where ATP is translocated to establish a endosymbiosis. The reason this is important, is because the entire introduction is written as a means of recapitulating the endosymbiosis process; however, the authors don’t provide sound rationale or direct precedence for assuming ATP translocation would be the mechanism for this unique evolutionary process. If this manuscript were written purely as an engineering paper, then I would not be as worried; however, this is not the case. From an engineering perspective, the results are interesting, as providing a new way to increase ATP availability may

have many applications. Unfortunately, the paper (specifically the Introduction) is currently written to have large evolutionary implications, which don't seem well supported.

I believe much of the confusion stems from the authors not directly stating that they are trying to complement mitochondrial endosymbiosis (which drives ATP translocation in eukaryotes), not plastid endosymbiosis (which drives C translocation). The authors need to fundamentally rewrite how they present the paper in the Introduction to clarify these points (rather than to just input a few sentences) to not mislead readers. There is a disconnect between the motivations set in the Introduction and the Experiments described. Specifically, increasing ATP translocation from a cyanobacterial endosymbiont is not teaching us about plastid endosymbiosis, because the original eukaryotic host already had a mitochondrion to provide ATP. Thus, I believe it is more appropriate to frame this study as an engineering study to see the implications of increasing ATP accumulation in the host when driven by photosynthesis.

We have now extensively altered our introduction and discussion to highlight that the engineering aspect of this study and its evolutionary implications. However, we still feel that there are significant evolutionary implications of this study which we have clarified in our introduction and discussion. (Introduction: see page 2, line 28 to page 3 line 27, Discussion: see page 9, lines 41-47, page 10 line 39 to page 11, line 8)

C assimilation is the one of the several important functions of present-day chloroplasts. Apart from C assimilation, modern day chloroplasts perform sulfate assimilation, nitrate assimilation, synthesis of amino acids, fatty acids amongst others. There is no direct evidence to suggest that C assimilation alone was the only key driver of early stages of endosymbiosis. And yes, there is no direct evidence to suggest that ATP was a key driver for endosymbiosis either. In fact, there is no direct evidence that conclusively determines what the exact drivers for cyanobacterial endosymbiosis were over a billion years back (citation 4). As mentioned above, we have now revised our introduction and discussion to highlight our engineering efforts and clarified the implications of this study on evolution of organelles, particularly chloroplasts. (Introduction: see page 2, line 28 to page 3 line 27, Discussion: see page 9, lines 41-47, page 10 line 39 to page 11, line 8)

Authors agree with the reviewer that there is robust evidence to suggest that eukaryotic host already had a mitochondrion to provide ATP when cyanobacterial endosymbionts were established. However, it is important to note that cyanobacteria facilitate light driven synthesis of ATP and O₂ using photophosphorylation mechanisms unlike the mitochondrial oxidative phosphorylation; this could have provided evolutionary advantage under certain conditions. It is even possible that this could have complemented mitochondrial ATP synthesis through oxidative phosphorylation. A statement describing this added to page 10, lines 45 to page 11, line 4 and citation 60 is added.

One reason there might be a limited number of doublings that ADP and Pi is running out in the cyanobacteria. Can the authors show this? If so, then the authors will have provided sufficient proof of the mechanism being ATP translocation. Is there a way to add excess ADP/Pi to the yeast to see if this supports further generations in the chimera?

Saccharomyces Cerevisiae are incompetent to perform direct translocation of ADP across cellular membranes, in fact this is the role of ATP/ADP translocases and transporters which are not localized to the yeast cellular membranes but are localized to the membranes of the intracellular organelles (for background literature, check citation 28, 29). We have also added citation 31 that describes the yeast ADP/ATP translocase localization to mitochondria.

BG-11 medium has Pi and this is supplemented to the growth medium as indicated in the methods section of the manuscript.

The DCMU studies are helpful in establishing the connection that the described phenomenon is photosynthetically driven. This is perhaps the single most convincing data of the manuscript. However, one question that the authors did not address in my first-round of reviewer comments was why the doublings in the are measured by cell count/spot on plates. It seems from Fig S10, it was possible to do liquid growth curves, so it's not clear why there isn't consistency in how growth was measured.

We observe most robust growth of the chimeras when plated on solid, photosynthetic selection medium, therefore, they were measure by cell count/spot. Note, as indicated in the figure legend, Fig. S10 corresponds to the effects of DCMU on yeast cells alone and cyanobacteria alone.

The authors say that they have added text describing the loss of endosymbionts, but all they have written is: "As per our genomic DNA analysis of the chimeras, after more than 20 rounds of doublings, we observe that the chimeras lose their viability possibly due to loss of cyanobacteria." This does not address my original concern that the authors have not looked at what is happening to the endosymbiont; rather, they are only speculating that there is a loss of cyanobacteria. Currently, the manuscript is still superficially descriptive. Although mechanistic understanding of what the observed transient growth in chimeras is not required for this to be a paper of interest, more thorough understanding of what is happening in this system is required for it to be more scientifically rigorous. The authors say that they are setting up a microscopy-based system to track live cells; however, this is beyond the scope of the manuscript. I agree this would be beyond the scope of the paper; however, my original comment was not to do single cell live tracking. I think providing wide-field micrographs at different time points to observe how the endosymbiont supposedly is lost over 20 doublings would suffice. This seems doable seeing that TIRF and confocal images could be collected for Figure 4.

We added: (i) additional widefield confocal images to Fig. S4 that show the tracking of cyanobacterial endosymbionts at various stages under selection conditions, where significant loss of cyanobacterial signals is seen at later stages; (ii) gel images that demonstrates the loss of cyanobacteria gDNA PCR signals during later rounds of growth (Fig. S11). These data confirm that yeast cells lose cyanobacterial signals after 20 doublings.

I believe that the claims of this manuscript are quite bold. The addition of the DCMU experiment assuages some of my concerns, but even just adding observational data of the endosymbiont of 20 doublings would provide much needed evidence that the system is functioning the way they describe it.

Necessary data is added (Fig. S4 and Fig. S11)

We thank Reviewer #1 for taking the time to provide critiques.

Reviewer #3 (Remarks to the Author):

Thank you for adding the TEM, I think this is important.

But, as expected, since the cyanobacteria are located in the cytoplasm, and since they have completely skipped phagocytosis, the discussion of SNAREs in the manuscript does not make sense. It is true, as the authors say, that "Literature suggests that one mechanism by which intracellular pathogenic bacteria avoid lysosomal degradation is through the expression of SNARE-like proteins, which are thought to inhibit SNARE-mediated membrane fusion through mimicry of the host SNAREs." But, as described in reference 38 cited by the authors, "Intracellular pathogens enter cells via phagocytosis. In their host cells, a significant number of intracellular microorganisms such as Mycobacterium, Salmonella, Chlamydia or Legionella, have the ability to create favorable intracellular compartments within which to multiply. This strategy relies on the capacity of bacteria to block undesirable fusion events with the cell's degradative compartments and their capacity to

create new advantageous fusion pathways with host vesicles."

Since these cyanobacteria have not entered the cell through phagocytosis, they have not had the opportunity to remodel their vacuolar compartment, and so while expression of SNAREs might be helping the stability of this system, it is quite unclear why it is helping, and it is likely that the reason is completely unrelated to the reasons SNAREs are implicated in pathogen survival.

This is a good point. While SNAREs are not necessary (Fig. 2A-D), they are certainly beneficial (Fig. 3). At this point it is not clear to us how the SNAREs provide advantage to synthetic endosymbiotic systems. We had previously observed similar benefits when we had engineered synthetic *E. coli* endosymbionts in yeast cells. We have added a couple of statements indicating this on page 5 line 28, and page 10, lines 26-31.

We thank Reviewer #2 for suggesting the TEM experiment, and thoughtful and insightful comments that helped us to improve our resubmission.

Reviewer #4 (Remarks to the Author):

General comments:

In this study, Cournoyer and colleagues constructed a hybrid represented by yeast cells containing an introduced autotrophic cyanobacterium *Synechococcus* PCC 7942. The main issue raised by Rev. 2 was the lack of evidence for the photosynthetic activity of the inserted cyanobacterium which should be essential for the growth of the hybrid cells. The revised version of the manuscript now provides several lines of evidence for this activity, which is summarized in the new paragraph "Growth dependence of the yeast/cyanobacteria chimeras on photosynthesis" on page 8. (i) Authors assessed the growth of the hybrid under day/night cycle and in the dark. Fig. 5 shows that light significantly accelerated growth of the organism. When the cyanobacterial cells were pre-adapted to the darkness before the fusion, the resulting hybrids did not grow in the dark at all. This can be considered rather convincing evidence for the photosynthetic activity of the cyanobacterium as the crucial factor for the hybrid survival. (ii) Addition of inhibitor of photosystem II DCMU into the growth medium resulted in a significant retardation of the hybrid light-dependent growth. (iii) Authors monitored fluorescence of cyanobacterial phycobilisome antennas and chlorophyll embedded within the cyanobacterial photosystems. This indicates an efficient energy transfer between phycobilisomes and photosystems indicating photosynthetic competence of the internalized cyanobacteria. Nevertheless, in the last remark of Rev. 2 time dependent fluorescence based assays at a single cell level were recommended as an excellent tool for evaluating the active photosynthesis (photosystem II). This would be, for instance, induction of fluorescence by illumination after previous dark adaptation, which unfortunately was not performed by the authors but would provide the convincing evidence for the photosynthetic activity in situ. Nevertheless, I consider the new experiments and their results sufficient for proving the need of cyanobacterial photosynthetic activity for growth of the yeast:Syn chimera, which was the main request of the Rev. 2.

Thanks, we are happy to know that the additional experiments helped us to resolve this concern.

An error for correction:

1. In the legend to Fig. S4 please correct:... fluorescence microscopy...

Thanks, corrected.

We thank Reviewer #4 for taking the time to read our resubmission.

Reviewers' Comments:

Reviewer #1:

Remarks to the Author:

I appreciate the authors making some changes to the introduction based on my comments. Although I don't entirely agree with the premise and motivations described in the background, the results are very interesting and are enough reason to have this work be published in order to foster future debate and conversation surrounding the implications of the work.

Reviewer #3:

Remarks to the Author:

We can agree to disagree about the SNARE wording. It's not my paper. It's a hard problem and the experiments are interesting, although a bit hard to understand in a cell biological context.

We thank the reviewers for taking the time to read our revised manuscript and providing critiques. Please see our responses below (in blue). In our responses, we have listed the changes we have made to our resubmitted manuscript.

REVIEWER COMMENTS

Reviewer #1 (Remarks to the Author):

I appreciate the authors making some changes to the introduction based on my comments. Although I don't entirely agree with the premise and motivations described in the background, the results are very interesting and are enough reason to have this work be published in order to foster future debate and conversation surrounding the implications of the work.

Following changes were made to introduction:

1. We have now further altered our introduction section to mainly focus on design and engineering aspect.
2. We also deleted a statement from our introduction that implies evolutionary implications of our work. (Following statement was deleted: "Given that *Synechococcus* are close relatives of chloroplast precursors, our observations may have significant implications on the early stages of evolution of photosynthetic eukaryotic life forms.")

Following changes were made to discussion:

1. In order to not overstate the evolutionary implications of our observations, the "evolutionary implications" section in the discussion was reworded and significantly shortened (in total, around 18 lines from this section were deleted)

We thank the Reviewer #1 for their time and efforts in helping us to improve our submission.

Reviewer #3 (Remarks to the Author):

We can agree to disagree about the SNARE wording. It's not my paper. It's a hard problem and the experiments are interesting, although a bit hard to understand in a cell biological context.

To avoid confusion about the hypothetical role of SNAREs in our observations and its evolutionary significance, we significantly altered the discussion section. Particularly, we removed the hypothetical role of SNAREs from the discussion section of the manuscript. We also deleted the part of the discussion section that implied the evolutionary significance of SNAREs.

We thank the Reviewer #3 for their time and efforts in helping us to improve our submission.